# Offline Hierarchical Reinforcement Learning via Inverse Optimization

**Carolin Schmidt**[1], **Daniele Gammelli**[2], **James Harrison**[3], **Marco Pavone**[2], **Filipe Rodrigues**[1]
[1]Technical University of Denmark, [2]Stanford University, [3]Google DeepMind
{csasc,rodr}@dtu.dk,{gammelli,pavone}@stanford.edu, jamesharrison@google.com

## Abstract

Hierarchical policies enable strong performance in many sequential decision-making problems, such as those with high-dimensional action spaces, those requiring long-horizon planning, and settings with sparse rewards. However, learning hierarchical policies from static offline datasets presents a significant challenge. Crucially, actions taken by higher-level policies may not be directly observable within hierarchical controllers, and the offline dataset might have been generated using a different policy structure, hindering the use of standard offline learning algorithms. In this work, we propose *OHIO*: a framework for offline reinforcement learning (RL) of hierarchical policies. Our framework leverages knowledge of the policy structure to solve the *inverse problem*, recovering the unobservable high-level actions that likely generated the observed data under our hierarchical policy. This approach constructs a dataset suitable for off-the-shelf offline training. We demonstrate our framework on robotic and network optimization problems and show that it substantially outperforms end-to-end RL methods and improves robustness. We investigate a variety of instantiations of our framework, both in direct deployment of policies trained offline and when online fine-tuning is performed. Code and data are available at `https://ohio-offline-hierarchical-rl.github.io`

## 1 Introduction

Deep reinforcement learning (RL) and optimal control (OC) have made significant progress within a broad range of continuous control tasks, such as locomotion skills (Lillicrap et al., 2015), dexterous manipulation (Zhu et al., 2019), and robotic navigation (Long et al., 2018). However, most of these tasks are inherently *atomic*, as they can be completed by performing basic skills episodically rather than through complex multi-level reasoning. Hierarchical policy decompositions, in which multiple sub-policies are composed to perform control at successively higher levels of temporal and representational abstraction, have long held the promise to help solve such complex tasks (Barto & Mahadevan, 2003; Nachum et al., 2018). Specifically, by defining a hierarchy of policies where higher levels influence the behavior of the lower levels, it becomes easier to train high-level policies to plan over longer time scales. Moreover, approaches for OC often leverage problem-specific structure by constructing hierarchical policies over convenient state representations, e.g., planning in operational space as opposed to direct joint space control for robot manipulation (Khatib, 1987; Peters & Schaal, 2008).

Similarly, sequential decision-making systems operating in the real world—such as vehicle routing and traffic control (Rasheed et al., 2020; Zardini et al., 2022), supply chain management (Rolf et al., 2023), and power grid optimization (Duan et al., 2019), among many others—have historically benefited from hierarchical abstractions. Hierarchically structured policies are commonly used to (i) decompose a large optimization problem into smaller, tractable ones (Fluri et al., 2019), and (ii) combine the benefits of differently-structured sub-policies, such as integrating learning-based methods with direct optimization (Delarue et al., 2020; Gammelli et al., 2023).

Nevertheless, there remains a significant gap between the theoretical promise of hierarchically structured policies and their practical application to complex, real-world decision-making problems: previous work often relies on costly online data collection, which is impractical for real-world, safety-critical systems. To address this issue, offline RL has gained attention for its ability to train policies from static offline datasets, thus avoiding the need for expensive or unsafe online exploration.

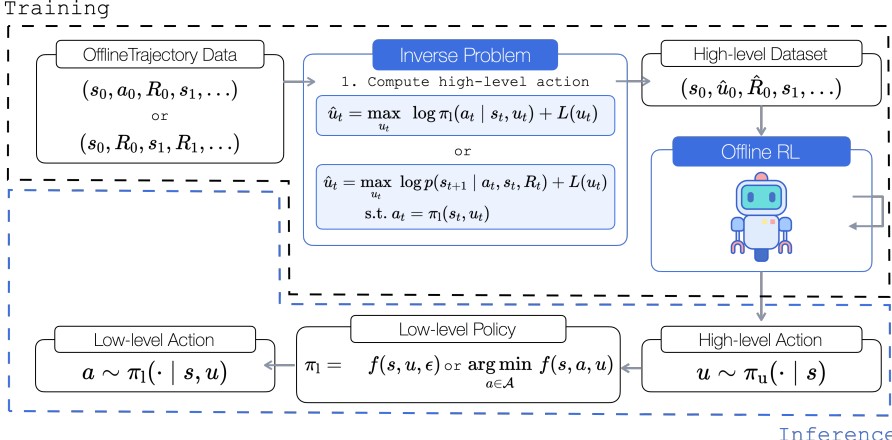

Figure 1: We propose *OHIO*, a framework to learn hierarchical policies from offline data. By exploiting structural knowledge of the low-level policy, we solve an inverse problem (top center) to transform low-level trajectory data (top left) into a dataset amenable to offline RL (top right), regardless of the nature of the policy used for data collection. At inference time, the RL-trained policy provides inputs to the low-level policy (bottom).

However, offline policy learning has had a limited impact within hierarchical formulations due to two fundamental issues. First, unless we assume that offline data collection is performed using the same hierarchical policy that we intend to learn, actions across hierarchy levels may not be observable, thus hindering the direct application of standard offline RL algorithms. Second, even if the offline dataset is collected using a hierarchical policy, any modifications to the hierarchical controller or its components can lead to ill-posed offline RL formulations.

In this work, we propose a framework to learn hierarchical behavior policies from offline data, *OHIO*: **O**ffline **H**ierarchical reinforcement learning via **I**nverse **O**ptimization. Specifically, we solve the *inverse problem*, mapping state transitions (and, optionally, low-level actions) to the high-level actions that likely generated those transitions. This approach allows us to create a dataset that can be used by standard offline RL algorithms within any hierarchical policy scheme, regardless of the nature of the policy used for data collection (Figure 1).

**Contributions.** Concretely, the contributions of this work are:

- We present a novel framework for hierarchical offline RL that leverages structural knowledge of the hierarchical policy to construct a dataset amenable to off-the-shelf offline RL algorithms.
- We derive principled inverse optimization objectives to solve the inverse problem both analytically and numerically, thus making our method amenable to generic policy structures and behavior policies used for data collection.
- We investigate design decisions and learning strategies within our framework, such as the impact of model learning, the choice of inverse optimization algorithm, dataset characteristics, and their impact on system performance and fine-tuning capabilities.
- Through experiments on robotic tasks, supply chain inventory control, and dynamic vehicle routing, we show how our framework substantially improves the performance of off-the-shelf offline learning algorithms across a diverse set of embodiments and policy structures, while providing the safety guarantees needed for safe, real-world deployment.

## 2 PROBLEM STATEMENT

We consider a discounted infinite horizon Markovian problem setting in which an agent interacts with a Markov decision process $\mathcal{M} = (\mathcal{S}, \mathcal{A}, P, r, \gamma)$. We denote the state and action at time $t$ as $s_t$, and $a_t$, and $\mathcal{S}, \mathcal{A}$ are the state and action spaces, respectively. Additionally, $P(s_{t+1} \mid s_t, a_t)$ denotes the (probabilistic) state transition dynamics, $r(s_t, a_t)$ denotes the reward function, and $\gamma$ is the discount factor. For brevity, we will also refer to single state transitions using $s$ and $s'$ for $s_t$ and $s_{t+1}$, respectively.

We will consider learning a hierarchical policy consisting of two components,

$$u \sim \pi_{\mathrm{u}}(\cdot \,|\, s), \quad a \sim \pi_{\mathrm{l}}(\cdot \,|\, s, u), \tag{1}$$

where $\pi_{\mathrm{u}}$ denotes a learned upper policy, and $\pi_{\mathrm{l}}$ denotes a fixed lower-level policy, and $u$ is the output of the upper policy that is an input to the lower-level policy. Our approach is generally agnostic to the form of the lower-level policy, although we will identify important example cases in the next section.

We focus on the case where the dataset comprises state-only trajectories, specifically considering $\tau_i = (s_0, s_1, ..., s_T)$ or $\tau_i = (s_0, r_0, s_1, r_1, ..., s_T)$. In this context, we assume access to approximate dynamics $\tilde{P}$ and reward models. These assumptions are reasonable, as our work targets common scenarios where certain elements of domain knowledge are inherently available. For instance, in robotic manipulation, having access to a robot arm's dynamics model is not only standard practice but essential for operating any low-level controller. Similarly, in network decision systems, real-world algorithms often depend on deterministic approximations of system dynamics. For example, the analysis of transportation systems frequently employs simple macroscopic models derived from traffic flow theory.

## 3 METHODOLOGY

In this section, we provide a comprehensive overview of the OHIO framework. We begin by presenting each component in a general setting, followed by specific examples of their implementations. The section begins with a discussion of the lower-level policies used and then moves on to address the inverse optimization problem and its corresponding solution methods. Lastly, we present the overall framework.

### 3.1 LOWER-LEVEL POLICIES

We will broadly consider two classes of lower-level policies: explicit and implicit policies. We distinguish between these two classes of policies, which are necessarily treated differently in policy inversion.

**Explicit policies.** (Stochastic) explicit policies require only a single function evaluation, or

$$a = f(s, u, \varepsilon), \tag{2}$$

where $\varepsilon$ is a generic random variable (enabling policy stochasticity through reparameterization) and $f$ is a generic function. This class of policies is broad. It includes policies that range from simple feedback controllers—such as PID, or linear state-space controllers such as those computed via LQR—to more complex policies such as those parameterized by neural networks and learned by RL or imitation learning (IL).

**Implicit policies.** Implicit policies, on the other hand, are defined as

$$\underset{a \in \mathcal{A}}{\arg\min} \quad f(s, a, u), \tag{3}$$

or approximate solutions thereof. This class includes optimization-based policies such as Model Predictive Control (MPC) (Rawlings & Mayne, 2013), and implicit learning-based policies such as diffusion-based or other implicit methods (Chi et al., 2023; Florence et al., 2022).

**Example 3.1** (Linear-quadratic-Gaussian). Let us consider a quadratic cost (negative reward) function with linearized Gaussian dynamics, where we assume the output of the upper policy, denoted as $\pi_{\mathrm{u}}$, is a goal post-transition state. Specifically, by Taylor-expanding the reward function (yielding terms $m$ and $M$) and approximate dynamics around the current state and zero action, the optimization problem becomes:

$$\begin{aligned} \underset{a \in \mathcal{A}}{\arg\max} \quad & -\frac{1}{2} \mathbb{E}_{s'} \left[ \|a - m\|_M^2 + \|s' - u\|_V^2 \right] \\ \text{s.t.} \quad & s' \sim \mathcal{N}(As + Ba + c, \Sigma), \end{aligned} \tag{4}$$

where $u$ is the goal state, $V$ is an arbitrary matrix that weighs satisfaction of achieving this goal state versus satisfying other reward terms[1], and $A, B, c$ are the terms resulting from Taylor-expanding the known dynamics. The solution to this problem is computationally tractable. In particular, if the action is unconstrained, this corresponds to a variant of the linear quadratic regulator (LQR) problem, where the optimal action is $a^* = Ku + k$ with policy parameters $K, k$ depending on the reward function and dynamics. [2]

---

[1]As has been noted by Gammelli et al. (2023), this term may be learned alongside the RL policy.

[2]We include the detailed derivation in Appendix A.3

## 3.2 POLICY INVERSION

Our framework aims to map available data to high-level action representations. In particular, we aim to compute high-level trajectories $\hat{\tau}_i = (s_0, u_0, r_0, ..., s_T)$ given a dataset of lower-level trajectories $\tau_i$.

**Lower-level policy inversion.** The core of our approach is to exploit the known optimization structure of the inner problem to approximately compute the most likely high-level action inducing each state transition and use these for offline training. We treat this problem as a probabilistic inference problem and aim to solve the regularized maximum likelihood estimation problem:

$$\hat{u}_{0:T-1} = \arg\max_{u_{0:T-1}} \quad \log \tilde{P}(s_{1:T} \,|\, a_{0:T-1}, s_{0:T-1}) + \sum_t L(u_t) \tag{5}$$
$$\text{s.t.} \quad a_t = \pi_1(s_t, u_t) \quad \forall t,$$

which is constructed jointly across each trajectory, and where $L$ denotes a regularization term. We require an approximate dynamics model $\tilde{P}$. In many applications, a simple approximate dynamics model, which only needs to be reasonably accurate for one timestep, is already known. This is often the case in robotics, where approximate models of the robot's dynamics are used for trajectory planning and motion control. In other scenarios, a dynamics model can be learned from the dataset. The use of an approximate dynamics model makes OHIO a novel combination of model-based and model-free RL. It leverages model information *locally in time*, while long-horizon performance is achieved through the RL-based component. This ensures the overall framework is not susceptible to compounding errors from multi-step prediction. When low-level actions are observed, as in the classical offline RL setting, we can bypass the need for an approximate dynamics model, as discussed in Appendix A.1.

In practice, we will decompose this joint likelihood across time and, assuming that the policy is stochastic, solve the one-timestep problem:

$$\hat{u}_t = \arg\max_{u_t} \quad \log \tilde{P}(s_{t+1} \,|\, a_t, s_t) + L(u_t) \tag{6}$$
$$\text{s.t.} \quad a_t = \pi_1(s_t, u_t).$$

In cases where the policy is deterministic and the likelihood under the policy is not well-defined, a simple alternative loss function (e.g., MSE loss) is used instead. This decomposition across time is sub-optimal, and the original joint-across-time problem has strong similarities to classical filtering and smoothing in partially-observed systems. However, we find that it is effective for our experiments due to the structure of many decision-making problems, leaving the consideration of the full joint likelihood for future work.

**Solving the policy inversion problem analytically.** How is this inverse problem solved? We will first illustrate one possible approach by building on the previous example.

**Example 3.2** (Analytical inverse: solving the inverse linear-quadratic problem). Here, we will continue Example 3.1, and discuss the solution of the inverse problem. Specifically, given the linearized Gaussian dynamics derived in Example 3.1:

$$\tilde{P}(s' \,|\, s, a^*) = \mathcal{N}(As + Ba^* + c, \Sigma), \qquad \text{for } a^* = \pi_1(s, u), \tag{7}$$

whereby $a^* = Ku + k$. The goal of the inverse problem is to compute the most likely high-level actions $u$ by solving Problem 6. By substituting $a^*$ into $\tilde{P}(s' \,|\, s, a^*)$, we obtain the following likelihood function describing the objective function for our inverse problem:

$$\mathcal{N}(As + BKu + Bk + c, \Sigma). \tag{8}$$

By expressing the likelihood in log terms and leveraging the fact that the log-likelihood is concave in $u$, we can easily derive its maximum value as:

$$\hat{u} = (BK)^{\dagger}(s' - (As + c + Bk)), \tag{9}$$

where $(\cdot)^{\dagger}$ denotes the Moore-Penrose inverse. We can then use $\hat{u}$, $a^*$, and $s$ to compute the reward $\hat{r}_t$. Ultimately, this leads to an analytical solution to the inverse problem via Equation 9.

This unconstrained linear-quadratic setting is one of the few that can be solved exactly. However, we emphasize an important distinction with work on differentiable optimization (Agrawal et al., 2019) and prior work on structured policies (Amos et al., 2018): our RL-based outer policy training does not require gradient propagation through the optimization problem, and thus any method may be used to solve the inverse problem. Indeed, because the inverse problem only needs to be solved to construct a dataset for offline training, comparatively expensive methods can be used.

---

**Algorithm 1** OHIO: Offline Hierarchical Reinforcement Learning via Inverse Optimization

---

**Require:** State transition dataset $\mathcal{D}$; Optionally: approximate dynamics $\tilde{P}$, reward function $r$.

  $\tilde{\mathcal{D}} \leftarrow \{\}$                                                     ▷ Initialize high-level dataset

  **for** $\tau \in D$ **do**

      Compute $\hat{\tau}$ via (6)                                  ▷ Low-level policy inversion

      $\tilde{\mathcal{D}} \leftarrow \tilde{\mathcal{D}} \cup \{\hat{\tau}\}$ ▷ If reward information is available, include in $\hat{\tau}$, otherwise compute $\hat{r}_t = r(s_t, \hat{a}_t)$

  **end for**

  Solve offline RL problem using $\tilde{\mathcal{D}}$ to **yield high-level policy** $\pi_{\mathrm{u}}$

---

**Solving the policy inversion problem numerically.** Several methods exist beyond analytical solutions, and our framework is agnostic to the method used. For discrete high-level actions, the inverse problem can be solved by exhaustive search over $u$. Similarly, we can employ sampling techniques for $u$ or zeroth-order optimization such as CEM (Rubinstein, 1997; De Boer et al., 2005). In certain cases, exact gradients may be computed through the inner problem, enabling the use of gradient-based optimization -such as gradient descent– to solve Problem 6. Thus, the numerical solution of the inverse problem is a considerably more general approach. More details are provided in Appendix A.4

### 3.3 FULL METHODOLOGY

Algorithm 1 highlights the relative simplicity of our approach: it focuses on leveraging approximate knowledge of the system dynamics and reward function to derive likely high-level actions and rewards. Subsequently, we apply off-the-shelf offline RL algorithms—including behavior cloning as a special case—on this dataset.

## 4 RELATED WORK

Our work is closely related to previous approaches to learning control within hierarchical policies (Ichter et al., 2018; Bansal et al., 2020; Xia et al., 2021; Lew et al., 2023) and offline RL within these settings (Le et al., 2018; Gupta et al., 2019; Zhou et al., 2021; Ajay et al., 2021; Rosete-Beas et al., 2023), providing a way to train general-purpose hierarchical policies from offline data.

**Hierarchical and Bi-Level RL.** Hierarchical IL jointly learns high-level policies and low-level controllers from optimal demonstrations (Le et al., 2018; Gupta et al., 2019). These methods have two main drawbacks: (i) they typically learn high-level actions in the form of sub-goals, thus, in the raw observation space, and (ii) they require oracle trajectory data. Our method alleviates both of these drawbacks by (i) learning high-level policies in an intermediate (potentially lower-dimensional) representation space, and (ii) leveraging offline RL methods to learn from sub-optimal data. Another class of methods uses offline RL to train the high-level policy in learned latent spaces (Zhou et al., 2021; Ajay et al., 2021). However, the policy used to generate the offline datasets may not match the hierarchical structure that we are interested in learning. Therefore, prior work typically formulates potentially complex, multi-step training schemes for the individual policy components, e.g., unsupervised trajectory autoencoders combined with hindsight relabeling to collect a dataset with the inferred high-level latent action and the respective reward (Rosete-Beas et al., 2023). To address these limitations, in our framework, we leverage approximate knowledge of the system dynamics to compute the most likely high-level action from raw trajectory data, thus avoiding the misalignment caused by intermediate objectives that do not necessarily correlate with the downstream task (e.g., reconstruction losses within generative models).

In robotics, numerous strategies have been developed for learning control with bi-level formulations that leverage traditional planning methods as inner components. For instance, prior work focuses on decomposing the overall policy into a high-level learned policy that generates waypoint-like representations for a low-level motion planner, e.g., based on sampling-based search (Ichter et al., 2018; Xia et al., 2021), model-based planning (Bansal et al., 2020), or trajectory optimization (Lew et al., 2023). Within this context, the high-level policy is typically learned through either imitation of oracle waypoint selection strategies or online RL. Analogously to previous methods, our approach uses the output of a higher-level, learned policy in a hierarchical structure. Crucially, however, we focus on solving complex control tasks from *offline data* by constructing datasets amenable to off-the-shelf offline RL algorithms.

**Offline RL and Learning from State-Only Demonstrations.** Lastly, our work is closely related to methods for learning from observations (LfO), by introducing a framework for offline RL from (potentially) state-only demonstrations. Distribution matching methods represent a principled approach to LfO (Boborzi et al., 2022), (Kim et al., 2022) by interactively estimating and minimizing the discrepancy between two stationary distributions: one generated by the expert and another by the learning agent. However, traditional approaches based on distribution matching typically require *online* interactions with the environment, with limited applications to tasks where exploration is expensive or unsafe. Moreover, methods that focus on learning from *offline* data typically cast LfO as an imitation learning problem, whereby the goal is to imitate the behavior of an expert policy and, thus, the overall performance can be limited by the quality of the data collection policy (Zhu et al., 2020b), (Qin et al., 2023), (Bewley et al., 2001). To address these limitations, our work introduces a new offline LfO approach to recover optimal policies from (potentially sub-optimal) state-only demonstrations.

## 5 EXPERIMENTS

In this section, we demonstrate the performance and broad applicability of our framework, OHIO, on two robotics scenarios (Section 5.1) and two real-world network optimization problems (Section 5.2). In particular, in the robotics scenarios, we evaluate a practical application of OHIO, where the high-level policy is learned through RL, and the low-level policy is an explicit policy, i.e. traditional, non-learned controller. In the network optimization scenarios, we demonstrate the performance of a particularly relevant instantiation of our framework, in which the low-level policy is optimization-based.

The goal of our experiments is to address the following key questions: (1) Can OHIO successfully recover hierarchical policies from datasets collected by arbitrary (i.e., non-hierarchical) behavior policies? (5.1, 5.2) (2) How do different inverse methods compare in performance? (5.1.1) (3) Does OHIO enable effective offline RL even when the dataset is collected with different or unknown low-level configurations? (5.1.2) (4) How does OHIO compare to traditional hierarchical RL? (5.1.2) (5) Does OHIO improve scalability and robustness relative to end-to-end approaches? (5.2)

**Benchmarking.** To isolate the contributions of OHIO, our analyses include the following comparisons: (i) OHIO with a known low-level policy, (ii) a traditional hierarchical RL approach with a known low-level policy but without the dataset reconstruction provided by OHIO (i.e., *"Observed State"* baseline, which selects the next observed state as the high-level action), (iii) a hierarchical RL formulation (i.e., *"HRL"*) in which both high-level and low-level policies are learned, and (iv) "flat" end-to-end approaches (i.e., *"End-to-End"*) with minimal architectural differences in the RL policy.

**Experimental design.** The learning algorithms used in this section include both off-the-shelf offline RL approaches (e.g., IQL (Kostrikov et al., 2021), CQL (Kumar et al., 2020)) and behavioral cloning (BC) algorithms. Dataset collection follows standard practices specific to each environment, such as (pre-trained) RL policies in robotics scenarios and domain-driven optimization or heuristic-based policies in network optimization. For consistent comparisons across datasets, we normalize scores to a range of 0 to 100, calculated as `normalized score` $= 100 * \frac{\texttt{score}}{\texttt{online RL performance}}$ (robotics) and `normalized score` $= 100 * \frac{\texttt{score}}{\texttt{oracle performance}}$ (network-optimization).

### 5.1 ROBOTICS

In this section, we focus on two distinct robotics scenarios: the first, traditionally solved using an end-to-end approach, is detailed in Section 5.1.1; the second scenario is typically addressed with a hierarchical reinforcement learning framework that includes non-learned lower-level controllers (i.e., RL policies guiding operational-space controllers for manipulation tasks), as discussed in Section 5.1.2.

#### 5.1.1 GOAL-DIRECTED CONTROL

We evaluate OHIO in a non-linear system using the *Reacher* task (Tunyasuvunakool et al., 2020), where the objective is to control a two-jointed robotic arm to move its end-effector to a randomly positioned target. This task is particularly suitable because it allows us to derive low-level policies with an analytical solution to the inverse problem, facilitating the comparison of different inverse methods—specifically, numerical approaches like gradient-based optimization versus analytical methods. Additionally, since this task is typically solved using end-to-end approaches, it serves as

Table 1: Normalized score on the reacher task comparing BC performance within End-to-End, "Observed State" and OHIO formulations, including the choice of algorithm for the inverse problem

| | OHIO | | | OBSERVED | |
| DATASET | NUMERICAL | ANALYTICAL | REG. ANALYTICAL | STATE | END-TO-END |
|---|---|---|---|---|---|
| HR DATASET | 97.1±10.2 | **98.2**± 3.2 | 97.1±10.2 | 0.05±0.0 | 95.4±14.2 |
| E2E DATASET | 99.2±15.0 | 94.8±22.0 | 95.4±23.4 | 0.04±0.0 | **99.3**±16.0 |
| E2E-10C DATASET | **95.3**±24.5 | 91.8±27.1 | 94.6±25.3 | - | - |
| E2E-10S DATASET | **98.4**±17.8 | 93.5±23.5 | 93.3±26.9 | - | - |

an effective demonstration of OHIO's capability to transform a "flat" (non-hierarchical) offline dataset into one suitable for offline hierarchical reinforcement learning.

**Datasets.** Our objective is to learn a policy from a dataset of non-hierarchical demonstrations (i.e. collected by an expert RL policy) using behavioral cloning (BC).

**The inverse problem.** We formulate the low-level policy as a linear feedback policy, specifically a finite-horizon LQR, where the high-level action is a goal state (position and velocity of robot joints). The detailed derivation of the inverse problem is provided in Appendix B.1.1.

**Choice of inverse algorithm.** We first evaluate OHIO under ideal conditions, on low-level demonstration data collected within the same hierarchical framework, i.e. a trained higher-level RL policy coupled with a lower-level LQR controller, and by assuming access to approximate dynamics for the inverse method. We refer to this case as "HR Dataset". Results in Table 1 demonstrate that OHIO can learn a policy closely matching the performance of the dataset. On the other hand, the baseline that selects the observed next state as the high-level action (i.e., "Observed State") results in an ineffective policy.

Access to the approximate (linearized) dynamics of the robotic arm is common practice and essential for operating the lower-level controller. However, we also investigate a more challenging scenario involving a dataset derived from demonstrations by an end-to-end agent (i.e., "E2E Dataset"). In this setting, we do not assume access to a dynamics model for policy inversion; instead, we learn an approximate model directly from the dataset. The results demonstrate that, despite the data being sourced from a different (i.e. flat) policy structure, OHIO achieves performance comparable to that of the end-to-end policy.

To evaluate OHIO's robustness to model misspecifications in the lower-level controller, we perturb the LQR parameters by increasing either the state or control cost by a factor of 10, resulting in datasets "E2E-10S" and "E2E-10C", respectively. The results in Table 1 reveal that while the analytical inverse is fast and exact, it is more susceptible to model misspecifications. Conversely, the numerical inverse maintains consistent performance across scenarios.

### 5.1.2 ROBOTIC MANIPULATION

Robotic manipulation tasks often benefit from integrating learning-based and non-learning-based components, making them an ideal domain for evaluating our proposed method. RoboSuite (Zhu et al., 2020a) is a widely used robotic manipulation environment that closely aligns with standard practices in real-world robotic implementations. In this setup, an RL agent interacts with a lower-level controller, abstracting away the direct control of joint torques.

**Datasets.** We generate datasets for two tasks within the RoboSuite environment:*Block Lifting* (i.e., "Lift") and *Door Opening* (i.e., "Door"). We follow a popular approach for offline RL data collection and utilize the replay buffers collected during online RL training. The default RoboSuite environment operates within a hierarchical framework, allowing for the collection of both high-level and low-level actions. This configuration enables us to evaluate OHIO's performance in reconstructing high-level actions in comparison to training on the original actions. Furthermore, we can assess OHIO's potential to facilitate effective offline RL under modified controller settings.

**The inverse problem.** We utilize the operational space controller of RoboSuite as our low-level policy, which computes the joint torques required to minimize the error between the current and goal pose (both position and orientation) of the end-effector. In this experiment, we use numerical inversion, showcasing the broad applicability of OHIO even in the absence of a closed-form solution for policy inversion.

**Robustness to low-level controller configurations.** The results presented in Table 2 highlight a fundamental advantage of OHIO over standard offline RL implementations (IQL). Traditional offline

Table 2: Normalized score comparing offline RL (IQL) to OHIO on robotic manipulation scenarios.

| | LIFT | | DOOR | |
|---|---|---|---|---|
| DATASET | IQL | OHIO | IQL | OHIO |
| ORIGINAL CONTROLLER | 88.5±20.7 | **89.6**±19.4 | 91.4±16.8 | **94.1**±14.1 |
| MODIFIED STIFFNESS | 86.8±16.4 | **98.9**± 4.4 | 18.6±14.3 | **92.7**±15.8 |
| MODIFIED DAMPING | 24.1±11.1 | **75.8**±30.5 | 2.9± 2.2 | **76.7**±28.2 |

Table 3: Normalized score comparing OHIO to offline hierarchical RL (HRL) with high-level goal in either (i) directly on the joint space, or (ii) same representation used by OHIO on robotic manipulation scenarios.

| TASK | OHIO | OBSERVED STATE | HRL - JOINT SPACE | HRL - REDUCED GOAL |
|---|---|---|---|---|
| LIFT | **89.6**±19.4 | 0.1±0.0 | 77.4±25.4 | 0.02±0.0 |
| DOOR | **94.1**±14.1 | 0.4±0.1 | 84.3±25.8 | 0.07±0.1 |
| DOOR - RED. DATA | **88.2**±25.4 | - | 72.7±34.8 | - |

RL tends to perform well when paired with the original controller used for data collection; however, its performance deteriorates significantly when the lower-level controller is configured with different parameters (IQL performance with modified stiffness and damping). In contrast, OHIO demonstrates strong performance across a wide range of parameters and tasks. Importantly, these findings demonstrate OHIO's potential to facilitate effective offline RL, even when data is collected using varied or unknown low-level configurations—a challenge that is exceedingly common in practical applications.

Moreover, results in Table 3 indicate that despite utilizing the same subgoal representation and lower-level policy as OHIO, the "Observed State" baseline struggles to learn the task effectively. This outcome clearly highlights the importance of the policy inversion method in enhancing performance.

**Choice of low-level policy.** Additionally, the results in Table 3 indicate that OHIO can improve the performance over HRL while allowing RL-based policies to be combined with standard low-level controllers. Decreasing the dataset size ("Door - Red. Data") reveals another advantage of OHIO: the performance of low-level action reconstruction is independent of dataset quality and coverage, whereas HRL shows a clear performance drop when not provided with an extensive dataset.

## 5.2 NETWORK OPTIMIZATION

In this section, we examine two real-world examples of societally-critical systems: *vehicle routing* and *supply chain management*. These problems represent real-world systems characterized by key features: (i) high-dimensional action spaces, such as nodes and edges in a large transportation network, (ii) complex system-level constraints that need to be strictly satisfied at all times (e.g., capacity limitations in warehouses), and (iii) readily available offline datasets of state transitions from system operators.

**Problem settings.** Vehicle routing problems are central to a wide range of mobility and logistics applications. The primary objective is to identify the least-cost routes for a fleet of vehicles while meeting the demands of geographically dispersed customers. In a similar vein, supply chain inventory management involves the strategic ordering and distribution of products within a network of interconnected warehouses and stores. The goal is to satisfy customer demand while minimizing system costs, which may include storage, transportation, and out-of-stock penalties, all while adhering to operational constraints like storage capacities. Comprehensive descriptions of the environments can be found in Appendices B.3 and B.4.

**Datasets.** To generate offline datasets, we simulate the operation of mobility-on-demand services and supply chains using both optimization and heuristic-based policies. In the vehicle routing scenario, we collect eight datasets across two real-world urban settings—NYC and Shenzhen—utilizing four different behavior policies: informed rebalancing ("INF") (Wallar et al., 2018), dynamic trip-vehicle assignment (DTV) (Alonso-Mora et al., 2017), a demand-proportional heuristic ("PROP"), and a random dispersion heuristic ("DISP"). For the supply chain scenario, we collect four datasets across two systems: one warehouse with three ("1W3S") and ten stores ("1W10S"), respectively. These datasets are generated using an optimization-based policy ("MPC") and a heuristic ("HEU") policy. We record the low-level actions, which represent the flows of vehicles or goods across the network.

Table 4: Normalized score comparing online (SAC) and offline (BC, IQL, CQL) algorithms within both End-to-End and OHIO formulations on the dynamic vehicle routing scenario.

| DATASET | BEH. POL. | END-TO-END | | | | OHIO | | | |
|---|---|---|---|---|---|---|---|---|---|
| | | SAC | BC | IQL | CQL | SAC | BC | IQL | CQL |
| NYC-INF | 98.5 ±1.7 | -35.2 ±8.3 | 88.7 ±1.5 | 48.2 ±1.3 | 48.1 ±1.5 | 98.0 ±1.9 | 97.6 ±2.3 | **98.1** ±2.8 | 93.0 ±1.7 |
| NYC-DTV | 89.4 ±2.1 | -35.2 ±8.3 | 67.0 ±1.6 | 48.9 ±1.5 | 69.2 ±2.3 | 98.0 ±1.9 | 89.2 ±2.3 | **91.1** ±2.8 | 83.5 ±2.3 |
| NYC-PROP | 85.7 ±1.5 | -35.2 ±8.3 | 83.1 ±1.7 | 42.2 ±1.6 | 68.3 ±1.8 | 98.0 ±1.9 | 85.7 ±2.5 | 85.8 ±2.2 | **88.0** ±2.4 |
| NYC-DISP | 45.8 ±0.7 | -35.2 ±8.3 | 44.1 ±2.7 | 57.4 ±2.1 | 32.5 ±2.9 | 98.0 ±1.9 | 86.5 ±1.6 | 82.8 ±2.2 | **94.1** ±1.7 |
| SHZ-INF | 90.9 ±0.7 | -7.7 ±3.3 | 90.1 ±1.5 | **90.4** ±1.4 | 42.2 ±1.4 | 95.5 ±1.0 | 87.0 ±1.0 | **90.4** ±1.4 | 88.8 ±1.6 |
| SHZ-DTV | 92.8 ±1.3 | -7.7 ±3.3 | 89.7 ±1.4 | 84.9 ±1.4 | 60.5 ±2.0 | 95.5 ±1.0 | **92.5** ±1.3 | 90.7 ±1.7 | 90.8 ±1.2 |
| SHZ-PROP | 84.5 ±1.0 | -7.7 ±3.3 | 85.5 ±1.4 | 86.5 ±1.2 | 59.8 ±1.6 | 95.5 ±1.0 | 83.3 ±1.0 | 83.6 ±1.1 | **87.4** ±2.3 |
| SHZ-DISP | 73.5 ±2.6 | -7.7 ±3.3 | 83.2 ±1.4 | **92.5** ±1.0 | 89.3 ±1.9 | 98.0 ±1.0 | 89.0 ±1.4 | **92.5** ±1.2 | 91.8 ±1.3 |

Table 5: Normalized score comparing online (SAC) and offline (BC, IQL, CQL) algorithms within E2E and OHIO formulations on the supply chain management scenario. ↓ refers to transfer performance between two environments (in this case, policies trained on 1W10S-MPC, tested on 1W10S-MPC-CAP).

| DATASET | BEH. POL. | END-TO-END | | | | OHIO | | | |
|---|---|---|---|---|---|---|---|---|---|
| | | SAC | BC | IQL | CQL | SAC | BC | IQL | CQL |
| 1W3S-HEUR | 81.7 ± 1.8 | 95.9 ± 2.4 | 80.3 ± 1.5 | 80.8 ± 1.7 | -203.6 ± 5.9 | 96.1 ± 2.0 | 79.4 ± 1.6 | **81.5** ± 1.5 | 79.0 ± 1.9 |
| 1W3S-MPC | 98.4 ± 1.8 | 95.9 ± 2.3 | 97.1 ± 2.4 | **97.6** ± 1.6 | -145.9 ± 2.6 | 96.1 ± 2.0 | 95.4 ± 2.0 | 96.0 ± 1.6 | 78.2 ± 1.9 |
| 1W10S-HEUR | 15.3 ± 3.0 | 87.5 ± 1.7 | -199.1 ± 146.7 | 4.54 ± 4.8 | -1220.3 ± 4.8 | 90.7 ± 1.1 | 10.9 ± 2.8 | 11.2 ± 4.1 | **13.4** ± 2.8 |
| 1W10S-MPC | 96.1 ± 1.4 | 87.5 ± 1.7 | 94.8 ± 1.0 | **95.1** ± 1.4 | -1677.8 ± 257.1 | 90.7 ± 1.1 | 91.8 ± 1.7 | 91.8 ± 1.8 | 6.0 ± 3.3 |
| | | | ↓ | ↓ | ↓ | | ↓ | ↓ | ↓ |
| 1W10S-MPC-CAP | 95.8 ± 1.3 | | 39.9 ± 33.3 | 45.8 ± 13.6 | -2110 ± 2.5 | | 86.7 ± 0.9 | **89.9** ± 1.4 | 32.2 ± 1.8 |

**The inverse problem.** We formulate the low-level optimization policies as linear programs (LPs), which allows us to exploit the fact that the inverse optimization problem of an LP can itself be formulated as an LP (Chan C. Y. et al., 2022). In practice, this results in an L1-norm minimization problem, thus projecting the low-level action onto the space of high-level actions within a feasible set of solutions. Specifically, the higher-level action commands a goal state representing a distribution of the commodities to be controlled (i.e. vehicles or goods). Please refer to Appendices B.3.5 and B.4.5 for a detailed derivation of the inverse.

**Scalability and robustness in direct deployment.** Results in Table 4 highlight a significant advantage of OHIO, which consistently outperforms E2E approaches. As observed in previous studies(Fluri et al., 2019; Gammelli et al., 2021; Skordilis et al., 2022; Singhal et al., 2024) E2E policies struggle with the high-dimensional action space inherent in large networks. Specifically, real-world transportation networks exhibit dense graph structures that result in an exponential growth of the (low-level) action spaces—196- and 289-dimensional for NYC and SHZ. In contrast, OHIO effectively capitalizes on the dimensionality reduction induced by the hierarchical decomposition, leading to 14- and 17-dimensional high-level action spaces for NYC and SHZ, respectively.

Similarly, Results in Table 5 highlight several important insights. First, OHIO enhances the performance of offline learning algorithms that require querying the value function on unseen actions during training (e.g., CQL). As noted in (Kumar et al., 2020), sample-based value estimation in high-dimensional action spaces poses significant challenges due to high variance and the curse of dimensionality. In this context, the hierarchical decomposition introduced by OHIO allows for more accurate value function estimation through dimensionality reduction, resulting in a more stable offline learning process.

Second, at first glance, there may not appear to be a clear advantage of OHIO when employing BC and IQL in the context of moderately sized graphs. However, the results in Table 5 reveal that E2E policies are *extremely* brittle, even when subjected to minimal variations in the scenario. Specifically, we evaluate both OHIO and E2E policies (trained on 1W10S-MPC data) in a minimally modified version of the same environment (i.e., 1W10S-MPC-CAP), where all state elements remain unchanged except for a reduction of storage capacity at store facilities from 15 to 10. Results in Table 5 illustrate the advantages of OHIO, with E2E policies experiencing a performance drop of at least 50%, whereas OHIO's performance only deteriorates by up to 5%.

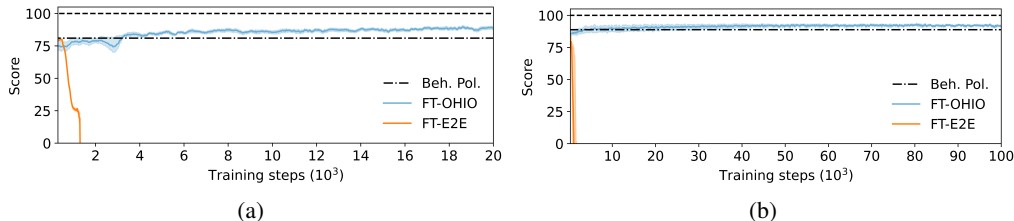

Figure 2: Supply chain fine-tuning performance of OHIO (FT-OHIO) and end-to-end (FT-E2E) policies pre-trained on (a) sub-optimal (i.e., HEUR) and (b) biased data (i.e., MPC with biased forecast).

**Robustness in online fine-tuning.** We further examine various scenarios of practical relevance, including fine-tuning policies that (i) are trained on sub-optimal data (Figure 2a), and (ii) must adapt to out-of-distribution bias (Figure 2b). Although both policies yield similar results during direct deployment (in the case of IQL), Figure 2 illustrates that OHIO policies demonstrate significantly greater stability and robustness during fine-tuning. The OHIO policy consistently improves upon the performance of the sub-optimal and biased policy it was trained on, whereas the E2E policy rapidly declines to a score below zero. Crucially, this degradation coincides with the E2E policy violating constraints during online interaction with the environment. This issue likely arises from two factors: (i) constraint violations are rarely included in offline datasets, as current operators must adhere to critical constraints during operations, and (ii) achieving hard guarantees within E2E architectures is challenging. In contrast, the OHIO policy can effectively encode domain-specific constraints through its low-level optimization-based policy, thereby avoiding infeasible out-of-distribution states by design.

Alongside the safety guarantees provided, this framework enables current system operators to train policies using offline data until they achieve a satisfactory level of performance. Subsequently, they can deploy these policies while safely enhancing performance through online interactions with the system.

## 6  DISCUSSION AND CONCLUSIONS

RL within large-scale, complex real-world systems has so far been limited by issues such as lack of robustness, sensitivity to distribution shifts, and expensive training processes. Hierarchical policy structures and offline RL are both promising strategies to tackle these issues, yet their integration remains an open challenge. To overcome the difficulties of combining offline RL with hierarchical policies, we propose an approach that leverages the structure of low-level policies along with approximate knowledge of the system dynamics and reward function. This approach formulates an inverse problem that transforms low-level state (and possibly action) information into datasets suitable for standard offline RL tools. OHIO not only successfully recovers hierarchical policies from datasets generated by arbitrary, i.e. flat behavior policies, but it also effectively utilizes datasets collected under varying or unknown low-level controller configurations—a common challenge in practice that often hinders the efficient use of available data (e.g., data collected across multiple robotic embodiments). Our approach demonstrates strong performance across all problem settings we evaluate, substantially outperforming end-to-end RL and other hierarchical approaches in terms of both performance and, crucially, robustness. While standard offline RL struggles to avoid constraint violations that are not present in the dataset, OHIO addresses this by directly encoding domain-specific constraints. As a result, OHIO inherently avoids infeasible out-of-distribution states, facilitating more robust deployment and safer online fine-tuning.

While our approach demonstrates considerable strengths, it also has certain limitations. Since OHIO integrates elements of both model-based and model-free reinforcement learning, its performance is sensitive to the accuracy of the dynamics approximation. Although we have not explored the robustness of our framework against model errors in this study, this represents a highly promising avenue for future research. Moreover, solving the inverse problem can be computationally intensive, even though this process is conducted entirely offline. In our current implementation, we simplified action reconstruction by neglecting temporal information for computational feasibility; however, more sophisticated estimation methods, such as cross-timestep losses, present a compelling direction for future exploration. More generally, we believe this research opens several promising directions for the extension of these concepts to a wider range of large-scale, real-world applications.

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

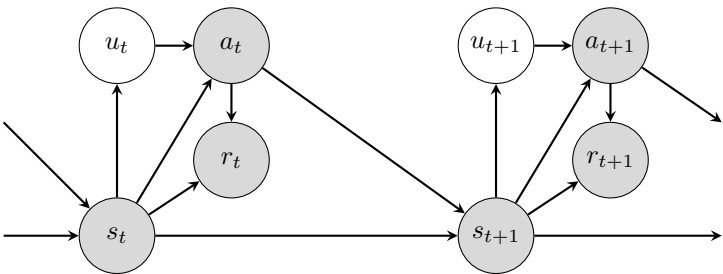

Figure 3: A graphical model for the system evolution, assuming Markovian dynamics and policies.

# A  ALGORITHMIC DETAILS

In this section, we provide further algorithmic details, and discuss practicalities of the policy inversion problem.

## A.1  POLICY INVERSION IN THE OBSERVED-ACTION CASE

We consider the case in which we assume access to a dataset $\mathcal{D} = \{\tau_i\}_{i=1}^N$ consisting of trajectories $\tau_i = (s_0, a_0, r_0, s_1, ..., s_T)$. This matches the information available in the classical offline RL setting; note that, critically, we do not assume access to upper-level actions $u$.

**Lower-level policy inversion.**  Our approach aims to recover the high-level actions that generated the observed low-level actions. We treat this problem as a probabilistic inference problem, and aim to solve the (regularized) maximum likelihood estimation problem

$$\hat{u}_{0:T} = \underset{u_{0:T-1}}{\arg\max} \quad \log p(a_{0:T} \,|\, u_{0:T}, s_{0:T}, r_{0:T}) + \sum_t L(u_t) \tag{10}$$
$$\text{s.t.} \qquad a_t = \pi_1(s_t, u_t) \quad \forall t,$$

which is constructed jointly across each trajectory, and where $L$ denotes a regularization term, e.g., $L_2$ regularization of the high-level actions. In practice, we will decompose this joint likelihood across time and (assuming that the policy is stochastic) solve the one-timestep problem,

$$\hat{u}_t = \underset{u_t}{\arg\max} \quad \log \pi_1(a_t \,|\, s_t, u_t) + L(u_t). \tag{11}$$

## A.2  POLICY INVERSION PROBLEM STATEMENT

The policy inversion problem is to recover $u_{0:T}$ from available data, provided in $\tau_i$ (corresponding to episode $i$), which includes either states, possibly actions, and possibly rewards. We include a graphical model for the system evolution in Figure 3, in the case where only the high-level actions are unobserved. In general, we may assume the low-level actions and/or the rewards are also unobserved.

There are numerous inferential procedures for the graphical model specified in Figure 3. In particular, EM-based methods or variational inference methods are possible to characterize uncertainty, and are well-established in e.g. hidden Markov models. In our settings, however, the mapping from high-level action to low-level action is typically underdetermined, and thus the inverse mapping is typically overdetermined. For example, in network control tasks, the high-level action corresponding to a goal state may be satisfied by many low-level (edge flow) actions.

Thus, we turn to a regularized maximum likelihood approach,

$$\hat{u}_{0:T} = \underset{u_{0:T}}{\arg\max} \quad \log p(\tau_i \,|\, u_{0:T}) + L(u_{0:T}) \tag{12}$$

which we decompose across time as previously mentioned, yielding inferential procedures shown in Figure 4.

In the action-observed case, it is sufficient to directly invert the policy. Typically, this takes the form of

$$\hat{u}_t = \underset{u_t}{\arg\max} \quad \log p(a_t \,|\, s_t, u_t) + L(u_t). \tag{13}$$

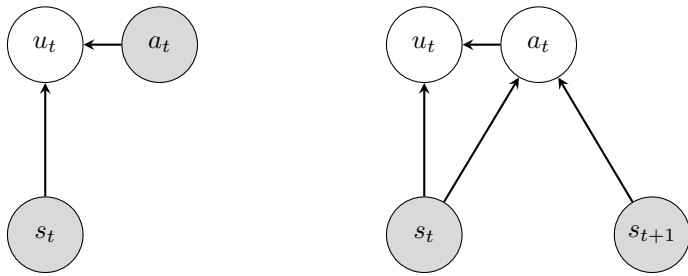

Figure 4: A diagram showing our inference procedure. Left: the observed-action case. Right: the case in which low-level actions are not observed.

In the case of explicit policies, this corresponds to a standard inverse problem, in which one aims to recover the input from an output. In some cases this is analytically tractable (as in the LQR example), but more commonly one must turn to numerical optimization, in which the objective corresponds to minimizing predictive error.

For implicit policies—especially optimization-based policies—this takes the form of inverse optimization (Chan C. Y. et al., 2022). Inverse optimization is analytically tractable in limited cases, but more generally one must turn to numerical methods.

When actions are not observed, the inversion procedure is similar to the action observed case. In contrast to e.g. EM procedures (which may be a natural inferential scheme to recover $a, u$ jointly) we first compute a point estimate of $a$ based on state transitions, and in turn use this to compute a point estimate $u$. While this may induce error in the inverse problem, we have found this scheme to work effectively in the settings we consider. Typically, systems dynamics are explicit—as opposed to optimization-based dynamics, which can occur for example in robotics with contact or multi-agent decision-making (Howell et al., 2022). Thus, if we define our dynamics as $s' = g(s, a)$, $\log p(s' \mid s, u)$ is straightforwardly written in terms of $a, u, s$, leading to a similar objective to the action-observed case.

### A.3 INVERSE PROBLEM: LINEAR-QUADRATIC GAUSSIAN

In this section, we provide the details on the linear-quadratic example from the body of the paper. Recall that the optimization policy is

$$\begin{aligned} \underset{a \in \mathcal{A}}{\arg\max} \quad & -\frac{1}{2}\mathbb{E}_{s'}\left[\|a - m\|_M^2 + \|s' - u\|_V^2\right] \\ \text{s.t.} \quad & s' \sim \mathcal{N}(As + Ba + c, \Sigma) \end{aligned} \tag{14}$$

where, by substituting, we have

$$\underset{a \in \mathcal{A}}{\arg\max} \quad -\frac{1}{2}\mathbb{E}_{s'}\left[\|a - m\|_M^2 + \|As + Ba + c + \epsilon - u\|_V^2\right] \tag{15}$$

for $\epsilon \sim \mathcal{N}(0, \Sigma)$. For the case where $\mathcal{A} = \mathbb{R}^{\dim(\mathcal{A})}$, this problem is concave, and thus for maximizer $a^*$,

$$0 = \mathbb{E}[M(a^* - m) + B^T V(As + Ba^* + c + \epsilon - u)] \tag{16}$$

$$= M(a^* - m) + B^T V(As + Ba^* + c - u) \tag{17}$$

which yields

$$a^* = (M + B^T V B)^\dagger[B^T V(u - As - c) + Mm] \tag{18}$$

which corresponds to $a^* = Ku + k$ for

$$K = (M + B^T V B)^\dagger B^T V \tag{19}$$

$$k = (M + B^T V B)^\dagger(Mm - B^T V(As + c)). \tag{20}$$

To compute the next state density, we can substitute this action, to yield policy density

$$s' \sim \mathcal{N}(As + B(Ku + k) + c, \Sigma) \tag{21}$$

which has a concave log-density. Again, the maximizer $u^*$ is achieved when

$$0 = (BK)^\top \Sigma^{-1}(s' - As + c + B(Ku + k)) \tag{22}$$

which is satisfied by

$$u^* = (BK)^\dagger(s' - (As + c + Bk)). \tag{23}$$

### A.4 Solving the Inverse Problem Numerically

We have shown that for a particular choice of inner policy and dynamics models, an analytical solution of the policy inversion problem is possible. While this approach is numerically efficient due to our recursive sensitivity calculation and our exploitation of problem convexity, we can instead turn to numerical solutions.

First, using automatic differentiation, the sensitivity of the low-level action (or post-transition state) to $u$ may be automatically computed. This enables a partially structured approach, in which we still exploit problem convexity, but automate computations that are potentially error-prone.

An alternative approach is simply turning to non-convex optimization methods—such as gradient descent—to compute an approximate minimizing $u$. Indeed, the only requirement for gradient descent is the differentiability of the policy with respect to $u$, as we have discussed previously. Finally, if the lower level is not differentiable we can resort to gradient-free methods, e.g. CEM. Thus, the numerical solution of the inverse problem is a considerably more general approach, and we will typically favor this approach. Algorithm 2 and Algorithm 3 illustrate the numerical inverse depending on available information in the dataset. Note, that in Algorithm 2, we could also calculate the loss over each state tuples in the lower-level (e.g. $\mathcal{L}(x_i, s_i)$), instead of only the resulting state.

---

**Algorithm 2** Numerical inverse - state-only trajectories

1: Given approximate dynamics $A, B$
2: Given $s, s_T \in \mathcal{D}$
3: $u \leftarrow s'$
4: **for** each step **do**
5:     Set $x_0 \leftarrow s$
6:     **for** $i = 0$ to $T$ **do**
7:         Get action $\hat{a} = \pi_l(x_i, u)$
8:         Unroll system dynamics: $x_{i+1} \leftarrow Ax_i + B\hat{a}$
9:     **end for**
10:    Compute loss $\mathcal{L}(x_T, s_T)$
11:    Update $u$ using $\mathcal{L}_{\text{total}}$, either using gradient descent or CEM
12: **end for**

---

**Algorithm 3** Numerical inverse with low-level actions $a$

1: Given, $s_{1..T}, a_{1..T}$
2: $u \leftarrow \vec{0}$
3: **for** each step **do**
4:     Initialize cumulative loss $\mathcal{L}_{\text{total}} \leftarrow 0$
5:     **for** $i = 0$ to $T$ **do**
6:         Get action $\hat{a}_i = \pi_l(s_i, u)$
7:         Compute loss $\mathcal{L}_{\text{total}} = +\mathcal{L}(\hat{a}_i, a_i)$
8:     **end for**
9:    Update $u$ using $\mathcal{L}_{\text{total}}$, either using gradient descent or CEM
10: **end for**

---

### A.5 Assumptions on Data Quality and Property

OHIO is agnostic to the learning signal used for training the high-level policy and, therefore, does not impose additional data quality assumptions beyond standard offline RL requirements. We demonstrate OHIO's effectiveness with both expert data -collected from trained RL agents and

traditional MPC policies- and suboptimal data—gathered from partially trained RL agents, heuristic- and optimization-based policies. We note that, when no approximate dynamics model is available, we require that low-level actions are observed to either learn an approximate dynamics model or directly perform policy inversion with low-level actions.

## B  EXPERIMENTAL DETAILS

In this section, we provide further detail about experiment details for the goal-directed control (Appendix B.1) and manipulation experiments (Appendix B.2). Further, we provide details on environment specifics relating to vehicle routing (Appendix B.3) and supply chain control (Appendix B.4) experiments, respectively and on learning components (Appendix B.5) for the network optimization tasks. The training of our models was executed on a Tesla V100 16 GB GPU.

### B.1  GOAL-DIRECTED CONTROL

For the first robotic experiment, we evaluate OHIO on the *Reacher-hard* task from the DeepMind Control Suite (Tunyasuvunakool et al., 2020). The end-to-end policy directly learns the low-level environment actions, whereas our hierarchical framework learns a desired goal state (position and velocity of robot joints), which serves as input to a goal-conditioned finite-horizon Linear Quadratic Regulator (LQR) with a horizon of $T = 5$. Specifically, the system is linearized at the current state using finite differences, yielding constant linear dynamics $A$ and $B$. The cost function penalizes deviations from the goal state, and we perform the Riccati recursion over a finite horizon $T$ to compute the time-varying feedback gains for control law $a = K(s - u)$. At each step of the recursion, the optimal feedback gain is computed as

$$K_t = -(R + B^\top P_{t+1} B)^{-1} (B^\top P_{t+1} A),$$
$$P_t = Q + A^\top P_{t+1}(A + BK_t).$$

To generate the datasets, we train both an E2E and a hierarchical SAC policy (Haarnoja et al., 2018) online and use the final checkpoint to collect the demonstration data. All policy and value function networks are MLPs with two hidden layers, each containing 256 units. Similarly, the learned dynamics model consists of two MLP layers with 256 units each and two output layers that map to the $A$ and $B$ dynamics matrices.

All datasets used for this experiment consist of 250 episodes of interactions (each consisting of 1000 timesteps). To learn the dynamics model, we use a train/val split of 0.9/0.1.

The SAC and BC algorithms use the following hyperparameters indicated in Table 6.

As lower-level policy we use an LQR policy with

$$Q = \begin{pmatrix} 10.0 & 0.0 & 0.0 & 0.0 \\ 0.0 & 10.0 & 0.0 & 0.0 \\ 0.0 & 0.0 & 1.0 & 0.0 \\ 0.0 & 0.0 & 0.0 & 1.0 \end{pmatrix}$$

and

$$R = \begin{pmatrix} 0.1 & 0.0 \\ 0.0 & 0.1 \end{pmatrix}$$

### B.1.1  ANALYTICAL INVERSE

Our goal is to compute the inverse analytically for an LQR policy that tracks goal state $u$ with a temporal abstraction $T$ to the higher-level policy.
First, without temporal abstraction, to compute the next state density, we substitute the action $a = K(s - u)$ to yield the policy density:

$$s' \sim \mathcal{N}(As + B(K(s - u)) + c, \Sigma), \tag{24}$$

which has a concave log-density. The maximizer $u^*$ is achieved when

$$0 = (BK)^\top \Sigma^{-1}(s' - As - B(K(s - u))). \tag{25}$$

This is satisfied by

$$u^* = (BK)^\dagger((A+BK)s - s').$$ (26)

We define two recursive terms $\Phi_1$ and $\Phi_2$ that evolve over time, allowing us to generalize our low-level policy across a temporal horizon $T$.

Initialization:

$$\Phi_1 = B_0 K_0, \quad \Phi_2 = (A_0 + B_0 K_0)s.$$

Recursive computation for $l = 1$ to $T$:

$$\Phi_1 = (A_l + B_l K_l)\Phi_1 + B_l K_l,$$
$$\Phi_2 = (A_l + B_l K_l)\Phi_2.$$

Final Solution for $u$:

$$u = -\Phi_1^\dagger(s' - \Phi_2).$$

**Regularised analytical inverse.** We show that using this analytical inverse formulation recovers the high-level action exactly in a linear state space model in Appendix C.1. However, in the main body, we apply this method to a more challenging, non-linear system using approximate linearized dynamics. When solving this inverse problem exactly, we observe large magnitudes in the recovered actions, as the solution attempts to perfectly fit the data under these approximate dynamics, which reduces generalizability and makes learning harder. To address this, we employ an implicit regularization technique by using the analytical gradient in a gradient descent algorithm, iteratively updating the solution with early stopping to prevent overfitting.

### B.1.2 NUMERICAL INVERSE

As an alternative to the analytical solution, we can solve the inverse numerically (see Algorithm 2). We use the Adam optimizer with a learning rate of $0.01$ and run 10000 steps per data point with early stopping if the difference between the previous and current loss is below $1e^6$. To avoid that solutions from bad local minima impact learning, we only include transitions with a loss $< 0.2$ in the dataset. Further, for all of the datasets we scale the action to be within $[-1, 1]$.

Table 6: Hyperparameters of SAC.

| Parameter | Value |
|---|---|
| Optimizer | Adam |
| Learning rate | $1 * 10^{-3}$ |
| Discount ($\gamma$) | 0.97 |
| Batch size | 100 |
| Entropy coefficient | 0.3 |
| Target smoothing coefficient ($\tau$) | 0.005 |
| Target update interval | 1 |
| Gradient step/env.interaction | 1 |

### B.2 ROBOTIC MANIPULATION

In Robosuite, we selected two tasks— *Door Opening* and *Lift* —that can be solved using online RL to collect datasets, both using standard environment configurations. We use the recommended lower-level controller, the Operational Space Controller, and set the temporal abstraction between the high-level RL and low-level controller to $T = 5$.

For data collection, we trained a standard online SAC (Haarnoja et al., 2018) algorithm for 1,500 episodes for Door Opening and 2,000 episodes for Lift to convergence. This process yielded datasets containing 750,000 transitions for Door Opening and 1,000,000 transitions for Lift, respectively. Data collection is done under standard controller settings with original stiffness ($kp = [150, 150, 150, 150, 150, 150]$) and damping ($kd = [1, 1, 1, 1, 1, 1]$). For the modified controller scenarios, we either change to $kp = [150, 150, 150, 50, 50, 50]$ or $kd = [3, 3, 3, 1, 1, 1]$.

For offline training of IQL, we use the parameters indicated in Table 7.

Table 7: Hyperparameters of IQL.

| Parameter | Value |
|---|---|
| Optimizer | Adam |
| Learning rate | $1*10^{-3}$ |
| Discount ($\gamma$) | 0.97 |
| Batch size | 256 |
| Target smoothing coefficient ($\tau$) | 0.005 |
| Target update interval | 1 |
| Gradient step/env.interaction | 1 |
| Temperature | 3 (Lift), 1 (Door) |
| Quantile | 0.7 (Lift), 0.9 (Door) |

### B.2.1 NUMERICAL INVERSE

For this lower-level controller, we resort to numerical methods. Fortunately, it remains differentiable, allowing us to utilize both gradient-based and gradient-free methods, such as the Cross-Entropy Method (CEM). In general we observe, that gradient-based approaches require fewer controller runs—thus reducing computational time—to converge to a solution. However, depending on the initialization of $u$, the higher exploration of CEM can help to escape local minima and lead to better results. (see Algorithm 3))

Consequently, in scenarios where we aim to restore the high-level action for the given controller, we initialize $u$ as a zero vector and run gradient descent. For transitions where gradient descent gets stuck in local minima, indicated by high final loss, we rerun CEM to improve the results.

In casesExec where we want to modify controller settings compared to the data collection, we can use the original high-level action as the initialization. In this case, gradient-based methods tend to perform well and deliver satisfactory results.

For the gradient descent algorithm, we use the Adam optimizer with a learning rate of 0.01, running for up to 10,000 steps with early stopping triggered if the loss falls below $1e^{-5}$. For the Cross-Entropy Method (CEM), we generate 50 samples per iteration, retaining the top 20% of them based on performance in each step. Early stopping is applied if the loss does not improve over the course of 4 consecutive steps.

### B.2.2 HIERARCHICAL RL ("HRL")

For the "HRL" baseline, where both levels are learned, we evaluate two scenarios: (i) the upper level predicts the same reduced goal state as in the case of an operation space controller (position and orientation of end-effectors) (ii) the upper level predicts the goal state directly in the low-level joint space (position and velocity of the robot joints. The lower-level network then outputs the required torques to reach the desired goal state. As the network architecture, we employ a recurrent neural network consisting of two LSTM layers and a fully connected MLP layer, followed by a Tanh activation function. The Tanh output is scaled to match the feasible range for torque control, ensuring the actions remain within valid limits. All hidden dimensions are set to 256, we use a train/val split of 09./01. and train for 100 epochs, where we save the model with the best validation loss.

### B.3 VEHICLE ROUTING

### B.3.1 ENVIRONMENT DETAILS

In our experiments, we focus on two case studies generated from trip record data, which we provide with our codebase, from the cities of New York, USA (NYC Taxi & Limousine Commission, 2013), and Shenzhen, China (Zhang et al., 2015). We are looking at a taxi-like system serving commute demand in the areas of Brooklyn and Shenzhen, respectively. In each scenario, the road network is segmented into geographical areas, representing stations. The trip record data are converted to demand, travel times, and trip prices between stations. As in (Gammelli et al., 2022), the arrival of passengers is assumed to be a time-dependent Poisson process, where the Poisson rate is aggregated from the trip record data every 3 minutes.

An on-demand service provider coordinates $M$ single-occupancy autonomous vehicles on a transportation network represented by a complete graph $\mathcal{G} = (\mathcal{V}, \mathcal{E})$ where $\mathcal{V} = \{v_i\}_{\{i=1:N_v\}}$ and $\mathcal{E} = \{e_j\}_{\{j=1:Ne\}}$ represent the set of vertices and edges of $\mathcal{G}$. Specifically, $\mathcal{V}$ defines the set of stations (e.g., pick-up or drop-off locations), and $\mathcal{E}$ defines the shortest paths between stations. The time horizon is discretized into a set of time steps $\mathcal{I} = 1, 2, ..., T$ of length $T$. At any time step $t$, vehicles are controlled to travel along the shortest path between station $i$ and $j \neq i \in \mathcal{V}$ with a travel time of $\tau_{i,j}^t \in \mathbb{Z}^+$ and travel cost $c_{ij}$, as a function of travel time. At each time step $t$, passengers submit trip requests for a desired origin-destination pair $(i,j) \in \mathcal{V} \times \mathcal{V}$, which is characterized by demand $d_{i,j}^t$ and price $p_{i,j}^t$. The operator matches passengers to vehicles, and the vehicles will transport the passengers to their destinations. For idle vehicles that are not matched with any passengers, the operator controls them to stay at the same station or rebalance to other stations. We denote $f_{ij,P}^t \in \mathbb{N}, f_{ij,P}^t \leq d_{i,j}^t$ as the passenger flow, i.e., the number of passengers traveling from station $i$ to station $j$ at time $t$ and $f_{ij,R}^t \in \mathbb{N}$ as the rebalancing flow, i.e., the number of vehicles rebalancing from station $i$ to station $j$ at time $t$.

### B.3.2 MDP DETAILS

*Reward $(r(s_t, a_t))$:* we choose the reward to be the operator's profit, which we define as the difference between the revenue from serving passengers and the cost of operations:

$$r(s_t, a_t) = \sum_{(i,j)\in\mathcal{E}} f_{ij,P}^{t+1}(p_{ij}^{t+1} - c_{ij}^{t+1}) - \sum_{(i,j)\in\mathcal{E}} f_{ij,R}^t c_{ij}^t.$$

*State space $(\mathcal{S})$:* the state space describes the current status of the transportation network via node features. Specifically, given a planning horizon $K$, we consider: (1) the current and projected availability of idle vehicles in each station $m_i^t \in [0, M], \forall i \in \mathcal{V}$ and $\{m_{i,j}^{t'}\}_{t'=t,...,t+K}$, (2) provider-level information trip price $p_{i,j}^t$ and cost $c_{i,j}^t$, (3) current $d_{ij}^t$ and estimated $\{\hat{d}_{i,j}^{t'}\}_{t'=t,...,t+K}$ transportation demand between all stations.

*High-level action $u$:* Given the number of idle vehicles and their current spatial distribution, we consider the problem of determining the *desired idle vehicle distribution* $q_i^{\hat{t+1}}$.

### B.3.3 MODEL IMPLEMENTATION

In what follows, we provide details for the implemented data collection methods and models.

**Optimization- and heuristic-based approaches:**

1. *Random Dispersion*: at each time step, the desired distribution is sampled from a Dirichlet prior with a concentration parameter $c = [1, 1, ..., 1]$.
2. *Informed rebalancing (INF)* (Wallar et al., 2018): This model assigns idle vehicles $\mathcal{V}_{\text{idle}}$ to rebalance to regions $\mathcal{R}$ that are reachable within a pre-defined time horizon $\mathcal{H}$. By including the forecasted demand rate in each region $\tilde{\lambda}_j$, it maximizes the expected number of requests each vehicle would observe in its assigned rebalancing regions. This formulation is extremely sensitive to the hyperparameters $\mathcal{H}$ and $\rho$. In our experiments, we tune them through an exhaustive grid search (Table 8, Table 9). Variable $y_{i,j}$ indicates assignment for vehicle $i \in \mathcal{V}_{\text{idle}}$ to rebalance to region center $j \in \mathcal{R}$ and $\tau_{i,j}$ gives the travel time from vehicle $i \in \mathcal{V}_{\text{idle}}$ to region center $j \in \mathcal{R}$.

$$\max \sum_{i\in\mathcal{V}_{\text{idle}}} \sum_{j\in\mathcal{R}} y_{ij} \cdot \tilde{\lambda}_j \cdot (\mathcal{H} - \tau_{i,j}) \tag{27a}$$

$$\text{s.t.} \quad \sum_{j\in\mathcal{R}} y_{ij} \leq 1 \qquad\qquad \forall i \in \mathcal{V}_{\text{idle}} \tag{27b}$$

$$y_{ij} \cdot (\mathcal{H} - \tau_{i,j}^t) \geq 0 \qquad\qquad \forall i \in \mathcal{V}_{\text{idle}}, j \in \mathcal{R} \tag{27c}$$

$$\sum_{i\in\mathcal{V}_{\text{idle}}} y_{ij} \cdot (\mathcal{H} - \tau_{i,j}) \leq \tilde{\lambda}_j \cdot \mathcal{H}^2 \cdot \rho \qquad\qquad \forall j \in \mathcal{R} \tag{27d}$$

3. *Dynamic trip-vehicle assignment (DTV)* (Alonso-Mora et al., 2017): This model assigns vehicles to unassigned requests by minimizing travel time and either assigning all idle vehicles

($\mathcal{V}_{\text{idle}}$) or all open requests ($\mathcal{R}_{ko}$). $y_{v,r}$ indicates assignment of vehicle $v \in \mathcal{V}_{\text{idle}}$ to request $r \in \mathcal{R}_{ko}$, while $\tau_{v,r}$ gives the shortest travel time for vehicle $v \in \mathcal{V}_{\text{idle}}$ to pickup up request $r \in \mathcal{R}_{ko}$.

$$\min_{\mathcal{Y}} \sum_{v \in \mathcal{V}_{\text{idle}}} \sum_{r \in \mathcal{R}_{ko}} \tau_{v,r} y_{v,r} \tag{28a}$$

$$\text{s.t.} \sum_{v \in \mathcal{V}_{\text{idle}}} \sum_{\mathcal{R}_{ko}} y_{v,r} = \min(|\mathcal{V}_{\text{idle}}|, |\mathcal{R}_{ko}|) \tag{28b}$$

$$0 \le y_{v,r} \le 1 \quad \forall v,r \in \mathcal{Y}. \tag{28c}$$

4. *Proportional Heuristic (PROP)*: This heuristic distributes excess vehicles according to the forecasted demand $\lambda_i$. It rebalances proportional to the averaged forecasted demand over the next $K = 6$ timesteps in region $i$ $\hat{q}_i = \frac{\lambda_i}{\sum_{j \in \mathcal{V}} \lambda_j}$

$$\min_{f_{ij,R}} \sum_{i \ne j \in \mathcal{V}} c_{ij}^t f_{ij,R} \tag{29a}$$

$$\text{s.t.} \sum_{j \ne i} (y_{ji}^t - f_{ij,R}) + q_i \ge \hat{q}_i, \, i \in \mathcal{V}, \tag{29b}$$

$$\sum_{j \ne i} f_{ij,R} \le q_i, \, i \in \mathcal{V}. \tag{29c}$$

Table 8: Hyperparameter tuning of SHZ-INF.

| $\mathcal{H}$ | $\rho$ | 1 | 2 | 3 | 4 | 5 | 6 |
|---|---|---|---|---|---|---|---|
| 2 | | 60.1 | 60.2 | 60.1 | 60.1 | 60.1 | 60.0 |
| 3 | | **60.9** | 59.6 | 59.6 | 59.8 | 60.0 | **60.1** |
| 4 | | 52.97 | 50.65 | 51.50 | 51.79 | 52.00 | 60.10 |
| 5 | | 50.83 | 44.62 | 44.19 | 44.19 | 44.19 | 52.10 |
| 6 | | 48.61 | 43.13 | 43.44 | 43.44 | 43.44 | 44.19 |

Table 9: Hyperparameter tuning of NYC-INF.

| $\mathcal{H}$ | $\rho$ | 1 | 2 | 3 | 4 | 5 | 6 | 7 |
|---|---|---|---|---|---|---|---|---|
| 2 | | 27.25 | 27.20 | 27.18 | 27.09 | 26.95 | 26.89 | 26.89 |
| 3 | | 39.70 | 38.30 | 39.17 | 39.30 | 29.30 | 39.30 | 39.30 |
| 4 | | 49.50 | 50.08 | 50.54 | 50.55 | 50.55 | 50.55 | 50.55 |
| 5 | | 52.88 | 54.21 | 54.23 | 54.23 | 54.23 | 54.23 | 54.20 |
| 6 | | 56.22 | 57.24 | **57.25** | **57.25** | **57.25** | **57.25** | **57.25** |
| 7 | | 56.68 | 56.04 | 56.04 | 56.22 | 56.22 | 56.22 | 56.20 |

**Learning-based approaches:**

1. *End-to-end RL*: for the end-to-end RL implementation, the flow action is defined along the edges as opposed to the desired distribution over nodes. We achieve this through minimal changes with respect to the OHIO network architecture. Specifically, this results in an edge convolution (consisting of 2 linear layers of 256 units) that outputs the mean and standard deviation parameters of a Gaussian policy for each edge in the graph.

2. *OHIO*: for all networks, we use one layer of GCN with 256 hidden units with a sum aggregation function, followed by 2 linear layers of 256 hidden units and a final layer mapping to the respective output's support.

### B.3.4 Optimization Policy Formulation

Given a desired next state described by the desired number of idle vehicles across stations $\hat{q}_i^t, \forall i \in \mathcal{V}$, we define the following linear control problem according to Gammelli et al. (2023) as follows:

$$\min_{f_{ij,R}^t} \sum_{(i,j)\in\mathcal{E}} c_{ij}^t f_{ij,R}^t \tag{30a}$$

$$\text{s.t.} \sum_{j\neq i}(f_{ji,R}^t - f_{ij,R}^t) + q_i^t \geq \hat{q}_i^t, \qquad i \in \mathcal{V} \tag{30b}$$

$$\sum_{j\neq i} f_{ij,R}^t \leq q_i^t, \qquad i \in \mathcal{V} \tag{30c}$$

$$f_{ij,R}^t \geq 0, \qquad (i,j) \in \mathcal{E} \tag{30d}$$

where the objective function (30a) represents the rebalancing cost, constraint (30b) ensures that the resulting number of vehicles is close to the desired number of vehicles, and with constraints (30c), (30d) ensuring that the total rebalancing flow from a region is upper-bounded by the number of idle vehicles in that region and non-negative.

### B.3.5 Analytical Inverse

We formulate the inverse problem using a data-driven inverse optimization approach Chan C. Y. et al. (2022). The general primal optimization problem models a network flow system that minimizes rebalancing costs and penalties for deviations from the desired target distribution (in cases where not all desired distributions are feasible and reachable). The primal problem is formulated as follows:

$$\begin{aligned} \text{Primal Objective:} \quad & \text{Minimize } c^\top x + p^\top z \\ \text{s.t.:} \quad & Ax + Bz \leq b, \\ & x \geq 0, \quad z \geq 0. \end{aligned}$$

,where in our example, $x$ denotes rebalancing flows, and $z$ is a slack variable penalizing deviations from the target distribution, with respective costs $c$ and $p$. The constraint matrix A encodes the flow and supply constraints, while B accounts for the deviations from the goal state.

In the inverse optimization problem, the goal is to reconstruct the unknown desired target distribution by adjusting the right-hand side (b) such that the observed solution ($x^*$) remains feasible and maximizes the fit of the forward model. This is achieved by minimizing absolute sub-optimality. The absolute sub-optimality problem can be reformulated and efficiently solved using strong duality Chan C. Y. et al. (2022).

$$\begin{aligned} \text{Minimize:} \quad & \epsilon \\ \text{s.t.:} \quad & Ax^* + Bz \leq b, \\ & A^\top \lambda \leq c, \\ & B^\top \lambda \leq p, \\ & \lambda \leq 0, \epsilon \geq 0, z \geq 0, b \in R \\ & \epsilon = (c^\top x^* + p^\top z) - b^\top \lambda, \end{aligned}$$

where specific components of $b$ are fixed to the current vehicle distribution. This ensures that the reconstructed $b$ aligns with the observed flows ($x^*$) while enforcing feasibility and minimizing the optimality gap under the adjusted model. The desired target distribution can then be inferred from the optimized $b$. We note, that this inverse model is bilinear, but Chan & Kaw (2020) show that this problem has an analytical solution. For increased exploration and robustness at the higher level, we could reconstruct different (potentially infeasible) target distributions that lead to the same observed flows. For this, we could add a small regularizer to the objective function, such as $\sum_r \alpha_r b[r]$, which encourages flexibility in reconstructing the target distribution, which is left as an interesting direction for future work.

On closer examination, in the current hierarchical formulation with action definitions on a fully connected graph — where all actions are feasible and reachable—, and since all the data collection strategies are central policies and we, therefore, observe optimal $x^*$, i.e. minimal cost flows, the inverse problem simplifies. Specifically, the constraint parameters that satisfy equation (30b) with equality $Ax^* = b$ provide one direct solution to the inverse problem. The reconstructed target distribution $\hat{q}_i^t$ is then computed as: $\hat{q}_i^t = q_i^t + \sum_{j \neq i}(f_{ji,R}^t - f_{ij,R}^t), i \in \mathcal{V}$.

## B.4 Supply Chain Inventory Management

In what follows, we describe environment specifics, MDP definitions, baseline implementation, and the low-level, optimization formulation.

### B.4.1 Environment Details

In our scenario, we consider a distribution network in a supply chain consisting of interconnected warehouses and stores aiming to meet customer demand while minimizing storage and transportation costs. We define the supply chain as a graph $\mathcal{G} = \{\mathcal{V}, \mathcal{E}\}$, where $\mathcal{V} = \mathcal{V}_d \cup \mathcal{V}_W$ is the set of warehouse $\mathcal{V}_W$ and distribution $\mathcal{V}_d$ nodes respectively and $\mathcal{E}$ the set of edges connecting warehouses to stores. If a sufficient inventory is available, demand $d_i^t$ is fulfilled in stores $s \in \mathcal{V}_d$ and sold at a price $p$. Unsatisfied orders are back ordered at a cost. At each time step $t$, warehouse $i$ orders additional units of inventory $w_i^t$ bounded by production capacity $C_p$ and stores available ones bounded by storage capacity $C_s$. Simultaneously, each store orders additional inventory from the warehouses bounded by storage capacity $c_i$. Ordered units get delayed by production $t^P$ and travel times $t_{i,j}$ respectively. During operations, production $m^O$, transportation $m^T$, storage $m^S$, and backorder costs $m^B$ occur. All stores are assumed to have an independent demand-generating process. We simulate seasonal demand behavior by representing demand $d_i \in \mathcal{V}_d$ as a co-sinusoidal function with a stochastic component defined as follows:

$$d_t^i = \left\lfloor \frac{d_{\max}^i}{2}\left(1 + \cos\left(\frac{f * \pi(2*r+t)}{T}\right)\right) + U(0, d_{\text{var}}^i) \right\rfloor,$$ (31)

where $d_{\max}^i$ is the maximum demand value, $U(0, d_{\text{var}}^i)$ is a uniformly distributed random variable, $T$ the episode length, $f$ and $r$ controlling the frequency and shift respectively.

Environment parameters are defined in Tables 10 and 11.

Table 10: Parameters for the 1F10S environment.

| Parameter | Explanation | Value | Parameter | Explanation | Value |
|---|---|---|---|---|---|
| $d^{\max}$ | Maximum demand | [5, 15, 20] | $m^S$ | Storage cost | [0.1, 0.5, 0.5, 0.5] |
| $d^{\text{var}}$ | Demand variance | [2, 2, 2] | $m^O$ | Production cost | 5 |
| $f$ | Demand frequency | [2, 4, 6] | $c_p$ | Production capacity | 25 |
| $r$ | Demand shift | [1, 3, 6] | $m^T$ | Transportation cost | 0.5 |
| $t_{ij}$ | Travel time | [1, 1, 1] | $p$ | Price | 15 |
| $t^P$ | Production time | 1 | $m^B$ | Backorder cost | 1.5 |
| $c$ | Storage capacity | [50, 15, 15, 15] | $T$ | Episode length | 30 |

Table 11: Parameters for the 1F10S environment.

| Param. | Explanation | Value | Param. | Explanation | Value |
|---|---|---|---|---|---|
| $d^{\max}$ | Maximum demand | [5,5,5,5,10,10,10,18,18,18] | $m^S$ | Storage cost | [0.005,2,…,2] |
| $d^{\text{var}}$ | Demand variance | [2, 2, 2] | $m^O$ | Production cost | 5 |
| $f$ | Demand frequency | [2, 4, 6, 2, 4, 6, 2, 4, 6, 3] | $c_p$ | Production capacity | 60 |
| $r$ | Demand shift | [1, 1, 1, 3, 3, 3, 6, 6, 6, 2] | $m^T$ | Transportation cost | 0.5 |
| $t_{ij}$ | Travel time | [1, 1, 1] | $p$ | Price | 15 |
| $t^P$ | Production time | 1 | $m^B$ | Backorder cost | 1.5 |
| $c$ | Storage capacity | [80,15,15,15,15,15,15,15,15] | $T$ | Episode length | 30 |

### B.4.2 MDP Details

*Reward* $(r(s_t, a_t))$: we select the reward function in the MDP as the profit of the inventory manager, computed as the difference between sales revenues and the sum of storage, production, transportation,

penalties for capacity violations, and backorder cost:

$$r(s_t,a_t) = \sum_{i \in \mathcal{V}_W} p \cdot \min(d_t^i, q_i^t) - \sum_{i \in \mathcal{V}} m_i^S \cdot q_i^t - \sum_{i \in \mathcal{V}_W} m_i^O \cdot w_i^t - \sum_{(i,j) \in \mathcal{E}} m_{ij}^T \cdot f_{ij}^t$$
$$- \sum_{i \in \mathcal{V}_d} 1.5 * p \cdot \max(0, q_i^t - c_s) - \sum_{i \in \mathcal{V}_d} m_i^B \cdot \max(0, d_i^t - q_i^t), \tag{32}$$

where $q_i^t$ is the inventory level at node $i$ at time $t$, $w_i^t$ the production order at warehouse $i \in \mathcal{V}_W$ at time $t$ and $f_{ij}^t$ the shipment flow from warehouse $i \in \mathcal{V}_d$ to store $j$ at time $t$.

*State Space ($\mathcal{S}$)*: the state describes the current state of the supply chain network by defining node and edge features. Node features contain (i) current and back-ordered demand, (ii) current inventory, (iii) storage and production cost, sales price, and storage and production capacities, (iv) incoming flow or orders for the next $T$ timesteps $\sum_{j \in \mathcal{V}} f_{ji:t+1:T}$ or $w_{i:t+1:T}$. Edge features are represented by the concatenation of (i) travel time $t_{ij}$ and (ii) transportation cost.

*Output of the RL policy $u$*: we define $u$ by two elements: (i) a goal production in warehouse nodes $w_i, \forall i \in V_w$ and (ii) a goal inventory over nodes $\hat{q}_i^t, \forall i \in V_d$.

### B.4.3 MODEL IMPLEMENTATION

In what follows, we provide additional details for the implemented dataset collection strategies and baselines.

**Domain-driven heuristics:**

1. *S-type Policy*: commonly known as the "order-up-to" policy, operates on the basis of the order-up-to level for the warehouses and stores. Essentially, at every time step the inventory manager places orders in an amount that will bring the total inventory on hand and in transit up to their respective order-up-to levels. In practice, the optimal order-up-to levels for each environment are determined through an exhaustive grid search.

**MPC-based:** Within this class of methods, we measure the performance of traditional optimization-based approaches using an MPC approach.

1. *MPC-Oracle*: this benchmark serves the purpose of quantifying the performance of an "oracle" controller. We provide this model with perfect foresight information on future demand and system dynamics. By providing the optimization model with Oracle knowledge of the realization of stochastic elements, we are able to quantify the optimality gap for the presented methods.
2. *MPC*: We relax the assumption of perfect foresight information in MPC-Oracle and substitute it with a noisy and unbiased estimate of demand.

**Learning-based:**

1. *End-to-end RL*: this benchmark does not approach the problem via the proposed hierarchical formulation of OHIO, but rather through more traditional end-to-end (E2E) architectures. Specifically, the flow action is defined along the edges as opposed to over the nodes. We achieve this through minimal changes to the architecture by an edge convolution (consisting of 2 linear layers of 32 hidden units) that outputs $\alpha$ and $\beta$ parameters of a Beta distribution for each edge in the graph. We adopt an individual upper bound for each action respective to the storage capacities/production capacities of the previous stage (Stranieri et al., 2024).
2. *OHIO*: for all networks, we use two layers of message-passing neural network of 256 hidden units with a sum aggregation function, followed by a linear layer mapping to the respective output's support.

### B.4.4 OPTIMIZATION POLICY FORMULATION

Given the output of the high-level RL-based policy defined as (i) the desired production $\hat{w}_i^t$ at warehouse nodes $i \in \mathcal{V}_W$ and (ii) the desired distribution of available inventory over distribution nodes

$\hat{q}_i^t, \forall i \in \mathcal{V}_d$, we define the following optimization-based policy Gammelli et al. (2023):

$$\min_{f_{ij}^t, w_i^t, \epsilon_{w,i}^t, \epsilon_{f,i}^t} \sum_{i \in \mathcal{V}_W} |\epsilon_{w,i}^t| + \sum_{i \in \mathcal{V}_d} |\epsilon_{f,i}^t| \tag{33a}$$

$$\text{s.t.} \sum_{j \in N^-(i)} f_{ji}^t = \hat{q}_i^{t+1} + \epsilon_{f,i}^t, \qquad \forall i \in \mathcal{V}_d \tag{33b}$$

$$\sum_{j \in N^-(i)} f_{ji}^t + q_i^t - d_i^t \leq C_{s,i}, \qquad \forall i \in \mathcal{V}_d \tag{33c}$$

$$\sum_{j \in N^+(i)} f_{ij}^t \leq q_i^t, \qquad \forall i \in \mathcal{V}_W \tag{33d}$$

$$q_i^t + w_i^t - \sum_{j \in N^+(i)} f_{ij}^t \leq C_{p,i}, \qquad \forall i \in \mathcal{V}_W \tag{33e}$$

$$w_i^t = \hat{w}_i^t + \epsilon_{w,i}^t, \qquad \forall i \in \mathcal{V}_W \tag{33f}$$

$$f_{ij}^t \geq 0, \qquad (i,j) \in \mathcal{E} \tag{33g}$$

where $w_i^t$ is the realised production at node $i \in \mathcal{V}_W$ at time $t$, $f_{ij}^t$ the commodity flows from node $i$ to node $j$ at time $t$, $d_i^t$ the demand in node $i \in \mathcal{V}_d$ at time $t$, $q_i^t$ the available inventory at each node $i \in \mathcal{V}$, and $C_{s,i}$ the storage and $C_{p,i}$ the production capacity at node $i \in \mathcal{V}$. The objective function 33a represents the distance metric that penalizes the deviation from the desired next states. Constraint 33b ensures that the total incoming flow in distribution nodes is as close as possible to the desired inventory, constraint 33c ensures that the inventory after demand realization, and incoming shipments do not exceed the storage capacity, constraint 33d guarantees that the combined shipment quantity is upper bounded by the warehouse inventory. Constraint 33e ensures capacity adherence in the warehouse, and constraint 33f ensures that orders from manufacturers are close to the desired orders quantity and, lastly, that commodity flows are defined as non-negative.

### B.4.5 ANALYTICAL INVERSE

The flow dynamics in this problem are deterministic, and the cost terms minimize the one-norm of error terms. These error terms measure disagreement between the realized next state under the system dynamics and the goal next state. In the context of inverse optimization, our goal is to infer the values of constraint parameters $\hat{q}_i^t \ \forall i \in \mathcal{V}_S$ (which correspond to high-level action u) that make the observed inventory flows optimal for the original LP. Similar to the vehicle routing scenario, we can achieve this through data-driven inverse optimization Chan C. Y. et al. (2022). When assuming that the observed flows do not violate capacity constraints, the direct solution to the inverse problem can be derived as $\hat{q}_i^{t+1} = \sum_{j \in N^-(i)} f_{ji}^t, \forall i \in \mathcal{V}_d$. Note, that this way we only generate feasible high-level actions for the offline dataset. The inclusion of error term $\epsilon_{f,i}^t$ in the inverse formulation to create non-feasible high-level action as means of exploration for the offline RL agent is left as future work.

### B.5 LEARNING COMPONENTS FOR NETWORK OPTIMIZATION

In this section, we provide details about the learning component in the network optimization experiments.

### B.5.1 NETWORK ARCHITECTURES

We parameterize policy, Q- and value function estimators through graph neural network encoders. The specific network architectures are problem-specific and can be summarized as follows:

1. Vehicle Routing: to define a valid vehicle distribution, the output of the policy network is sampled from a Dirichlet distribution $u_t \sim \text{Dir}(u_t|c)$. More precisely, we use a Graph convolutional neural network (GCN) (Kipf & Welling, 2017) with sum aggregation function, followed by three linear layers that compute the concentration parameters $c \in \mathbb{R}_+^{N_v}$ over $N_v$ regions, where the positivity of $c$ is ensured by a Softplus nonlinearity. The Q- and value functions have the same backbone architecture. In the Q-function architecture, the node

Table 12: Hyperparameters of SAC.

| Parameter | Value |
|---|---|
| Optimizer | Adam |
| Learning rate | $1*10^{-3}$ |
| Discount ($\gamma$) | 0.97 |
| Batch size | 100 |
| Entropy coefficient | 0.3 |
| Target smoothing coefficient ($\tau$) | 0.005 |
| Target update interval | 1 |
| Gradient step/env.interaction | 1 |

     embeddings are concatenated with the action before being fed into the linear layers. The value function maps from node embeddings to the final value estimate through a sum aggregation.

2. Supply Chain Inventory Management: we use a message-passing neural network (MPNN) (Gilmer et al., 2017) with sum aggregation. The output of the policy network is defined as (i) concentration parameters $c \in \mathbb{R}_+^V$ of a Dirichlet distribution over warehouses and stores for computing the shipment flows, and (ii) $\alpha \in \mathbb{R}_+^{|\mathcal{V}_W|}$ and $\beta \in \mathbb{R}_+^{|\mathcal{V}_W|}$ of a Beta distribution, where the output is scaled by the production capacity to define the desired production. The Q- and value functions share the same encoder architecture. The action is concatenated with the node embeddings before the linear layers to achieve a Q-value estimate, while the node embeddings are aggregated through summation to compute the value function estimate.

### B.5.2 ONLINE FINE-TUNING

In this section, we state the specifics of our online fine-tuning procedure.

With offline RL, we obtain a policy initialization, which is intended for sample-efficient online fine-tuning. Offline learned policy and value functions via IQL or BC can be fine-tuned directly. However, conservative methods such as CQL tend to learn smaller Q-values than their true values. Consequently, initial interactions during online fine-tuning are spent adjusting the Q-function, leading to unintentional unlearning of the initial policy. To address this, we calibrate the Q-function during offline learning to match the range of the ground-truth Q-values via Cal-CQL, as proposed in Nakamoto et al. (2023).

Further, to mitigate large gradient updates during initial fine-tuning that potentially lead to unstable policy behavior, we (i) freeze the weights of the policy network and only train the Q- and/or value function for the first 200 episodes (ii) we start by sampling $50\%$ of the batch from the offline dataset and gradually decrease this proportion to 0 over 3000 episodes.

### B.5.3 HYPERPARAMETERS

**SAC:** The hyperparameters used to train our online baseline can be found in Table 12. To improve learning stability, we implement (i) a double estimator for the Q-function (Hasselt et al., 2010) and (ii) target Q-networks (Mnih et al., 2015).

**CQL:** For our offline experiments, we train the $CQL(\mathcal{H})$ version of CQL with trade-off factor $\alpha = 1$. We use a policy learning rate of $1*10^{-4}$ and a critic learning rate of $3*10^{-4}$. The remaining hyperparameters are kept identical to the online SAC version in Table 12.

**IQL:** We use $\tau = 0.9$ and $\beta = 3$ for all implementations and a policy learning rate of $1*10^{-4}$ and a critic learning rate of $3*10^{-4}$. General hyperparameters are kept the same as in the online SAC version in Table 12.

## C  FURTHER EXPERIMENTAL RESULTS

### C.1  ANALYTICAL INVERSE ON LINEAR STATE SPACE MODEL

Given a linear state space model with state transition matrix $\mathbf{A}_s$ and control input matrix $\mathbf{B}_s$: $\Delta t = 0.5$,
$\mathbf{A}_s = \begin{bmatrix} 1 & \Delta t \\ 0 & 1 \end{bmatrix}$ and $\mathbf{B}_s = \begin{bmatrix} \frac{\Delta t^2}{2} \\ \Delta t \end{bmatrix}$, we recover several high-level actions with Equation (23) for different
LQR parameter settings that exactly lead to the next observed state, which are illustrated in Figure 5.
Specifically, we test $R \in \{0, 0.2\}$, and $Q_{11} \in \{3.5, 4.0, 4.5, 5.0, 5.5\}$ and $Q_{22} = 1$, $Q_{12} = Q_{21} = 0$.

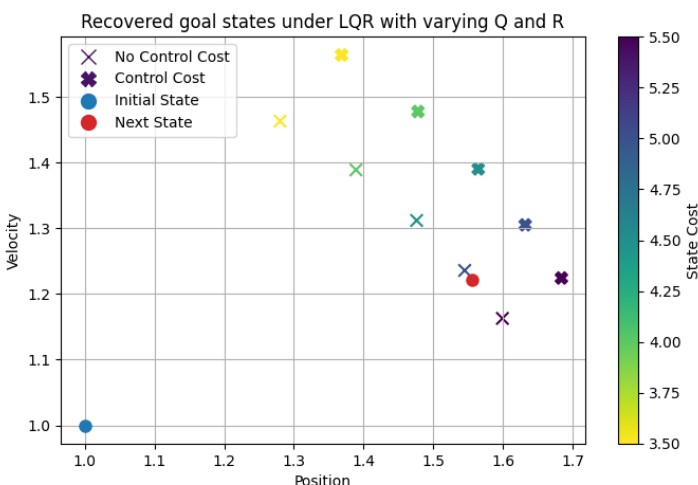

Figure 5: Visualistion of high-level actions recovered by the analytical inverse for different LQR parameter settings

### C.2  VISUALIZATION OF POLICIES IN ROBOTIC MANIPULATION

We provide a visualization of the policies obtained by standard offline RL in Figure 6a), which fails to perform the task of opening the door, and OHIO Figure 6b) performing effective offline RL and completing the task.

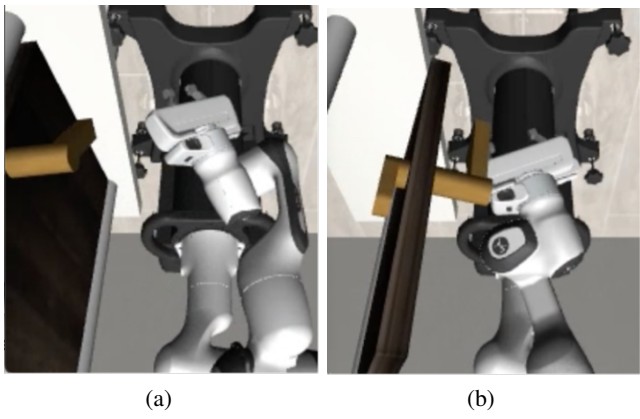

(a)  (b)

Figure 6: Door opening task with modified stiffness (a) standard offline RL (No Sucess), (b) OHIO (Succes)

### C.3 LEVERAGING DIFFERENT DATA SOURCES FOR ROBOTIC MANIPULATION

We introduce an "Expert Dataset" comprising 250 episodes collected using an expert policy for the Door Opening task. Using this dataset, we compare:

- i) Hierarchical imitation learning (HIL). This approach applies an imitation learning objective to both the high- and low-level policies. The high-level policy predicts a goal state in the joint space (position and velocity of the robot joints), whereas the low-level learns the robot torques to reach this goal state.
- ii) OHIO-based Imitation Learning ("OHIO-IL"). This method employs an imitation learning objective for the high-level policy, combined with the inverse optimization process derived by an operational space controller for the low-level policy. The high-level policy predicts a goal state in the operational space (position of the robot end effectors).

As shown in Table 13 (left) learning solely from expert demonstrations proves challenging due to limited data coverage, which impacts both HIL and OHIO-based methods and prevents them from consistently solving the task. To address this limitation, we leverage a broader set of demonstrations to enhance offline learning. Specifically, we construct a "Combined Expert Dataset" by merging demonstrations from three distinct expert policies, each collected under a different controller setup (200 episodes per policy, totaling 600 episodes)—a setup commonly encountered in practice. As shown in Table 13 (right), OHIO demonstrates its effectiveness in integrating diverse data sources, outperforming non-OHIO-based methods. Furthermore, offline RL objectives enable learning from datasets created by multiple expert policies, consistently outperforming imitation learning approaches (e.g., HIL vs. HRL and OHIO-IL vs. OHIO-IQL).

Table 13: Normalized scores on the door opening task

| Expert Dataset | | Combined Expert Dataset | | | |
|---|---|---|---|---|---|
| HIL | OHIO - IL | HIL | HRL | OHIO - IL | OHIO - IQL |
| **57.73** ±39.2 | 26.44 ±37.9 | 29.89 ±34.8 | 64.75 ±37.7 | 81.85 ±33.1 | **91.1** ±23.2 |

### C.4 COMPARISON OF SAMPLE-EFFICIENCY DURING ONLINE TRAINING

We provide the training curves in Figure 7 to demonstrate the significant improvement in sample-efficiency and learning stability of hierarchical RL over traditional E2E RL.

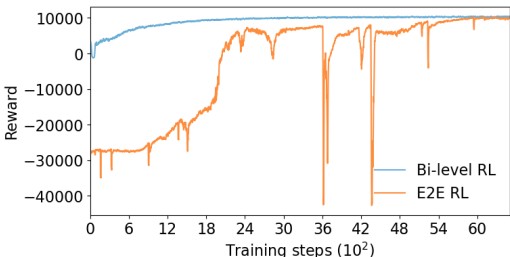

Figure 7: Training curve for online hierarchical (Bi-level) RL and E2E RL in the supply chain 1W10S experiment.

### C.5 COMPARISON OF RUNTIME AT INFERENCE

We perform additional analyses on run times for the DVR problem (Table 14). We show how, despite the substantially increased graph sizes, the computation time of our method remains tractable across different real-world scale.

Table 14: Runtimes of OHIO at inference on the Dynamic Vehicle Routing environment on different graph sizes.

| No. Edges | 16,000 | 10,000 | 40,000 | 90,000 |
|---|---|---|---|---|
| E2E | 0.02 s (± 0.04) | 0.04 s (± 0.00) | 0.16 s (± 0.01) | 0.32 s (± 0.01) |
| OHIO | 0.09 s (± 0.01) | 0.73 s (± 0.01) | 5.61 s (± 0.04) | 14.90 s (± 0.25) |
| MPC-Oracle (T=6) | 0.47 s (± 0.02) | 3.93 s (± 0.38) | 21.97 s (± 2.58) | 44.13 s (± 2.76) |
| MPC-Oracle (T=12) | 1.69 s (± 0.28) | 45.21 s (± 3.5) | 85.23 s (± 7.92) | 163.06 s (± 11.29) |

### C.6 ONLINE FINE-TUNING OF OHIO POLICY IN NETWORK OPTIMIZATION SCENARIOS

We further evaluate the performance of the policy learned by OHIO during online fine-tuning, both in-distribution (i.e., within the same city) and in a transfer learning setting (i.e., requiring adaptation to a different city, with unseen topology, demand patterns, travel times, etc.). Results in Figure 8 show how, in both cases, OHIO policies are able to reliably improve upon the starting performance learned from offline data. Crucially, the policy learned by OHIO is *consistently above the performance of the behavior policy* during the entire fine-tuning process, thus avoiding prohibitively expensive low-reward interactions during the initial phases of training and potentially alleviating a critical bottleneck for the deployment of RL within real-world systems. In Figure 9, we show how the E2E policy is unable to adhere to constraint violations during online fine-tuning.

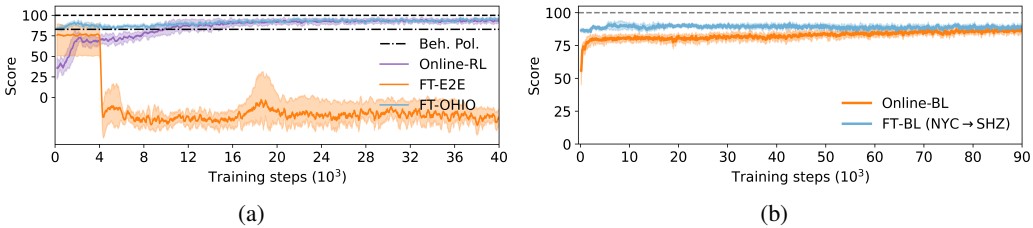

(a)          (b)

Figure 8: Vehicle routing fine-tuning performance (y-axis) of pre-trained hierarchical policies (FT-OHIO) compared with training from scratch (Online-BL) as a function of gradient steps deriving from online interaction (x-axis) with either (a) a same-city scenario (NYC→NYC) or (b) in a transfer learning setting (NYC→SHZ). "Beh. Pol." indicates the performance of the behavior policy.

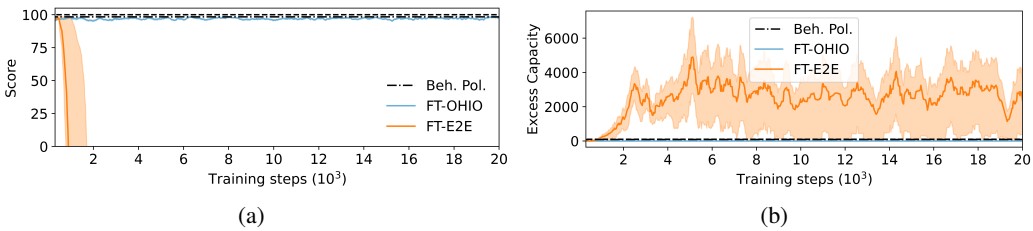

(a)          (b)

Figure 9: Supply chain fine-tuning performance (a) and constraint violation (b) of OHIO (FT-OHIO) and end-to-end (FT-E2E) policies pre-trained on near-optimal data (i.e., MPC).

