# OpenReview forum: "Offline Hierarchical Reinforcement Learning via Inverse Optimization"
_ICLR.cc/2025/Conference — ICLR 2025 Poster_

### Official Review · Reviewer_dYhu · 2024-11-01

**Soundness:** 3
**Presentation:** 3
**Contribution:** 3
**Rating:** 5
**Confidence:** 3

**Summary:**

This paper presents OHIO, a framework for offline reinforcement learning (RL) using hierarchical policies. The framework uses inverse optimization to recover high-level actions that generated the observed data. The authors perform experiments in robotic manipulation and network optimization environments, and show improvements against end-to-end and hierarchical policies.

**Strengths:**

1. The paper presents an interesting approach for effectively leveraging the inherent hierarchical structure for downstream learning.

2. The authors show impressive results on roobotic and network optimization environments, with many ablations pertaining the hierarchical structure.

3. It covers good related work section and the limitations of this method are clearly mentioned in the discussion section.

**Weaknesses:**

1. The main weaknesses in this paper are the experiments. Although the authors mention prior related hierarchical approaches for offline learning, the comparisons with such sophisticated approaches are missing in the experiment section. This makes it hard to judge the efficacy of the method against prior approaches. A more detailed comparison with such baselines on complex multi-level reasoning tasks is thus necessary.

2. Further, Algorithm 1 provides minimal information for the hierarchical formulation, and can be significantly improved by adding appropriate details of the training procedure. Extensive details on how the hierarchical levels interact with each other during training is also missing. The paper would benefit by adding pseudocode for how the high-level and low-level policies are trained and interact, with details on how the inverse optimization is integrated into the training loop.

3. Minor: Typo in Line 201.

**Questions:**

Can you provide comparisons of OHIO with prior hierarchical approaches that leverage offline RL, e.g OPAL: Offline Primitive Discovery for Accelerating Offline Reinforcement Learning [1] and approaches that leverage expert demonstrations like Relay Policy learning [2].

Can you discuss how would the method compare against prior methods that leverage expert demonstrations for learning the inherent hierarchical structure, e.g. Relay Policy learning [2].

[1]. OPAL: Offline Primitive Discovery for Accelerating Offline Reinforcement Learning (Anurag Ajay, Aviral Kumar, Pulkit Agrawal, Sergey Levine, Ofir Nachum)
[2]. Relay Policy Learning: Solving Long-Horizon Tasks via Imitation and Reinforcement Learning (Abhishek Gupta, Vikash Kumar, Corey Lynch, Sergey Levine, Karol Hausman)

---

> ### Author Response · Authors · 2024-11-21
>
> We thank the reviewer for their comments. Below, we respond to each comment/question individually.
>
> ***Algorithm 1 provides minimal information for the hierarchical formulation, and can be significantly improved by adding appropriate details of the training procedure. Extensive details on how the hierarchical levels interact with each other during training is also missing. The paper would benefit by adding pseudocode for how the high-level and low-level policies are trained and interact, with details on how the inverse optimization is integrated into the training loop.***
>
> We thank the reviewer for the opportunity to clarify OHIO's training process. Specifically, the original manuscript outlines OHIO’s sequential approach in Algorithm 1, which consists of two key stages:
> - *High-Level Dataset Generation*: Prior to offline training, a high-level dataset is constructed from low-level trajectories by solving the inverse problem. This step is performed once for the entire dataset.
> - *High-Level Policy Training*: Using the high-level dataset generated by the inverse problem in Step 1, the high-level policy is trained offline with any standard offline RL algorithm.
>
> Algorithm 1 was designed to provide a high-level overview of the OHIO framework rather than to introduce specific mathematical formulations or training routines. This design choice reflects OHIO’s versatility, as it is agnostic to the specific inverse optimization strategies, low-level policy structures, or training methodologies used.
>
> Regarding the reviewer’s comment on low-level policy training, it is important to note that during the inverse optimization process in OHIO, there is no interaction between the low-level and high-level policies. Instead, the structure of the low-level policy is embedded in the definition of Equation (6). As such, low-level policy training is outside the scope of Algorithm 1. Further, while our framework is amenable to various kinds of low-level controller, we do not directly incorporate learned controllers in this work.
>
> In light of this discussion, and motivated by the reviewer’s suggestions, we have refined Algorithm 1 in the revised manuscript to enhance clarity and explicitly address these points.

---

> ### Author Response · Authors · 2024-11-21
>
> ***Can you provide comparisons of OHIO with prior hierarchical approaches that leverage offline RL, e.g OPAL: Offline Primitive Discovery for Accelerating Offline Reinforcement Learning [1] and approaches that leverage expert demonstrations like Relay Policy learning [2].***
>
> We thank the reviewer for the pointers to additional methods for offline learning of hierarchical policies. In what follows, we discuss approaches that leverage expert demonstrations [2]. and offline RL objectives [1] in order.
>
> **Leveraging expert demonstrations**:
>
> Methods that rely on imitation learning, such as [2], are closely related to our HRL benchmark. Both approaches use imitation learning to train the low-level policy, but they differ in how the high-level policy is trained. Specifically, [2] employs imitation learning objectives to imitate expert policies, whereas our HRL benchmark uses offline RL objectives to learn from diverse—and potentially suboptimal—policies.
>
> Motivated by the reviewer's suggestion, we further investigate the role of expert demonstrations by introducing additional baselines and purposefully selected datasets. Specifically, we introduce an “Expert Dataset” comprising 250 episodes collected using an expert policy for the Door Opening task. Using this dataset, we compare:
> - Hierarchical imitation learning (HIL). This approach, similar to [2], applies an imitation learning objective to both the high- and low-level policies. The high-level policy predicts a goal state in the joint space (position and velocity of the robot joints), whereas the low-level learns the robot torques to reach this goal state as in [2].
> - OHIO-based Imitation Learning (“OHIO-IL”). This method employs an imitation learning objective for the high-level policy, combined with the inverse optimization process derived by an operational space controller for the low-level policy. As in Section 5.1.2, the high-level policy predicts a goal state in the operational space (position of the robot end effectors).
>
> As shown in Table 1, consistent with the findings in [2], learning solely from expert demonstrations proves challenging due to limited data coverage, which impacts both HIL and OHIO-based methods and prevents them from consistently solving the task.
>
> *Table 1: Normalised scores on the door opening task - Expert Dataset*
> | **HIL**     | **OHIO - IL** |
> |:-----------:|:-------------:|
> |  **54.79**  |    32.94  |
>
> To address this limitation, we leverage a broader set of demonstrations to enhance offline learning. Specifically, we construct a “Combined Expert Dataset” by merging demonstrations from three distinct expert policies, each collected under a different controller setup (200 episodes per policy, totaling 600 episodes)—a setup commonly encountered in practice.
> As shown in Table 2, OHIO demonstrates its effectiveness in integrating diverse data sources, outperforming non-OHIO-based methods. Furthermore, offline RL objectives enable learning from datasets created by multiple expert policies, consistently outperforming imitation learning approaches (e.g., HIL vs. HRL and OHIO-IL vs. OHIO-IQL).
>
> *Table 2: Normalised scores on the door opening task - Combined Expert Dataset*
>
> | **HIL**     | **HRL**       | **OHIO - IL** | **OHIO - IQL** |
> |:-----------:|:-------------:|:-------------:|:--------------:|
> |   34.13 |    64.75  |    88.85  |    **90.45**  |

---

> ### Author Response · Authors · 2024-11-21
>
> We continue on our previous comment:
>
> **Alternative Hierarchical RL formulations**:
>
> While we agree with the reviewer on the relevance of works like [1] in the broader hierarchical RL literature, we want to highlight a fundamental difference between  [1] and OHIO. Specifically, [1] is based on latent goal selection, while methods like OHIO and other baselines implemented in our work are based on physically-interpretable goals. Crucially, by expressing the high-level policy over learned latent spaces, the policy used to generate the offline datasets may not match the hierarchical structure that we are interested in learning (e.g., a high-level policy interfacing with a low-level controller in operational space). Therefore, prior work, such as [1], typically formulates potentially complex, multi-step training schemes for the individual policy components, e.g., unsupervised trajectory autoencoders combined with hindsight relabeling to collect a dataset with the inferred high-level latent action and the respective reward.
> In our work, we purposefully avoid learned latent spaces and compute the most likely high-level action from raw trajectory data, thus avoiding the misalignment caused by intermediate objectives that do not necessarily correlate with the downstream task (e.g., reconstruction losses within generative models).
> Most importantly, a key strength of our framework lies in its ability to leverage non-learning-based controllers, which are by far the most commonly used in the application areas we consider (e.g., operational space controllers for manipulation tasks). This is a distinctive feature of OHIO, making direct comparison with methods like [1] less informative.
> In light of this, we hope the reviewer agrees with us in deeming explicit comparisons with methods like [1] an interesting—albeit, orthogonal—direction for future work.
>
> In conclusion, with the addition of these experiments, the revised manuscript now incorporates a comprehensive set of comparisons designed to systematically isolate the individual contributions of specific components of OHIO. Specifically, we implement: (i) OHIO with a known low-level policy, (ii) hierarchical RL with a known low-level policy but without the OHIO dataset reconstruction (“Observed State”), (iii) hierarchical RL with both high-level and low-level policies learned, (iv) hierarchical imitation learning from expert demonstrations (as in [2]), and (v) multiple “flat” RL and IL approaches.
>
> [1]. OPAL: Offline Primitive Discovery for Accelerating Offline Reinforcement Learning (Anurag Ajay, Aviral Kumar, Pulkit Agrawal, Sergey Levine, Ofir Nachum) [2]. Relay Policy Learning: Solving Long-Horizon Tasks via Imitation and Reinforcement Learning (Abhishek Gupta, Vikash Kumar, Corey Lynch, Sergey Levine, Karol Hausman)

---

> > ### Comment · Reviewer_dYhu · 2024-11-25
> >
> > I appreciate the author's responses and additional experiments which have further clarified the position of the proposed approach with respect to expert demonstrations based approaches.
> >
> > While I agree with the authors that approaches like OPAL learn latent subgoals which is different from the proposed approach framework, I still believe that missing comparisons with current state-of-the-art HRL approaches is an important limitation with the paper in its current form. Further, there are many HRL approaches like [1], [2] which learn physically interpretable subgoals and the proposed approach needs to be compared with such prior HRL approaches. Therefore, I have decided to maintain my original score.
> >
> > [1] Kim J, Seo Y, Shin J. Landmark-guided subgoal generation in hierarchical reinforcement learning[J]. Advances in neural information processing systems, 2021, 34: 28336-28349.
> >
> > [2] Wang, Vivienne Huiling, et al. "State-conditioned adversarial subgoal generation." Proceedings of the AAAI conference on artificial intelligence. Vol. 37. No. 8. 2023.

---

> > > ### Author Response · Authors · 2024-11-26
> > >
> > > We thank the reviewer for acknowledging the additional experiments provided during the rebuttal phase. Below, we address the reviewer’s comments in detail and clarify why the suggested comparisons may not be directly applicable to the context of OHIO.
> > >
> > > While we share the reviewer’s ambition for including as many comparisons as possible, we believe it is equally important to prioritize comparisons that offer meaningful insights to readers and, more broadly, to the ML community.
> > >
> > > Specifically, we appreciate the reviewer’s pointers to related work but feel the need to emphasize that the approaches mentioned (i.e., [1] and [2]) focus exclusively on **online HRL**. As such, they are unlikely to yield additional insights in the context of **offline HRL** methods like OHIO. In particular:
> > >
> > > - [1] aims to improve exploration via improved subgoal selection, relying on dynamically identified landmarks tied to the online exploration process, which are inherently absent in offline settings.
> > > - [2] addresses non-stationarity in the online training of high- and low-level policies by utilizing current online data from the low-level policy to train a discriminator. This approach also presupposes interaction with the environment, which is unavailable in offline scenarios.
> > >
> > > In other words, both methods are specifically designed for online HRL scenarios and cannot be meaningfully compared to offline HRL methods, where training occurs without environmental interaction. For this reason, we believe comparisons with these works would not provide significant additional value in the context of our study.
> > >
> > > Beyond the two works mentioned by the reviewer, we acknowledge the existence of offline HRL approaches that learn physically interpretable subgoals, such as IRIS [3] and POR [4]. However, the exclusion of these methods from the set of benchmarks was intentional, as they require low-level policies to be learned. Specifically, these approaches focus primarily on improving learning of high-level policies, while the low-level policy is typically trained via supervised learning using the offline dataset (similar to the HRL and IL baselines in our work). Notably, the high-level policy designs introduced by IRIS and POR could be integrated into OHIO in a modular manner, making these methods complementary rather than directly comparable with each other.
> > >
> > > Furthermore, a core contribution of our approach lies in its ability to integrate non-learned low-level controllers. This capability is particularly important in scenarios where constraints or domain-specific requirements are more effectively encoded within non-parametric controllers. In contrast, existing HRL methods, including [1-4], typically assume fully learned hierarchical policies. As a result, they face limitations similar to those observed in the E2E and HRL baselines in our experiments. For example, these methods struggle to incorporate domain-specific constraints, which non-learned controllers can naturally handle (e.g., as observed in the network optimization experiments).
> > >
> > > We genuinely hope this explanation addresses the reviewer’s concerns and provides clarity on why the suggested comparisons are not directly applicable to our framework. We sincerely appreciate the reviewer’s feedback and hope that our responses can be a starting point for further discussion on the relevance of additional baselines within the time limits of this rebuttal phase.
> > >
> > > [1] Kim J, Seo Y, Shin J. Landmark-guided subgoal generation in hierarchical reinforcement learning[J]. Advances in neural information processing systems, 2021, 34: 28336-28349.
> > >
> > > [2] Wang, Vivienne Huiling, et al. "State-conditioned adversarial subgoal generation." Proceedings of the AAAI conference on artificial intelligence. Vol. 37. No. 8. 2023.
> > >
> > > [3] Mandlekar, Ajay et al. “IRIS: Implicit Reinforcement without Interaction at Scale for Learning Control from Offline Robot Manipulation Data.” 2020 IEEE International Conference on Robotics and Automation (ICRA)
> > >
> > > [4] Xu, Haoran, et al. "A policy-guided imitation approach for offline reinforcement learning." Advances in Neural Information Processing Systems 35 (2022)

---

> ### Comment · Reviewer_dYhu · 2024-11-27
>
> I acknowledge that the suggested baselines consider online HRL setting. However, there are other baselines which employ offline HRL (as mentioned by the authors), or that use behavior priors at the lower level [1,2]. Thus, instead of pointing out the framework differences with prior approaches, I encourage the authors to pick various relevant state-of-the-art approaches of their choice, and demonstrate the efficacy of their approach. Also, it is a good research practice to re-implement the prior baselines according to the considered framework, while ensuring fair comparisons.
>
> [1] Avi Singh et al: Parrot: Data-Driven Behavioral Priors for Reinforcement Learning
> [2] Karl Pertsch et al: Accelerating Reinforcement Learning with Learned Skill Priors

---

> ### Author Response · Authors · 2024-11-29
>
> We thank the reviewer for their prompt response and deeply appreciate the valuable feedback provided throughout this rebuttal phase. Your efforts to engage with our work have been immensely helpful in refining our contributions.
>
> We would also like to take this opportunity to clarify our intentions regarding the previous response. From the outset, extensive evaluation and fair comparisons with alternative methods have been central to this work. This commitment is demonstrated by the 14 baselines implemented across the 7 experimental settings included in the original submission. Inspired by the reviewer’s comments, we have worked to the best of our capabilities to further implement additional baselines within the limits of the rebuttal phase, as demonstrated by the results of the new baseline and additional ablation on the use of offline RL included in our previous response.
>
> At the same time, we recognize the importance of balancing the inclusion of relevant methods with ensuring that comparisons remain both meaningful and insightful. Because of this, a significant portion of our efforts during this rebuttal phase has been dedicated to carefully assessing whether certain methods provide fair and insightful comparisons to those already included in the manuscript.
>
> As a result of balancing these two objectives—comprehensively addressing state-of-the-art methods for offline HRL while maintaining clarity and comparability among baselines—we chose to prioritize baselines that align with these criteria while transparently discussing our rationale for excluding others.
>
> Importantly, the arguments presented in our previous response stemmed from a careful and collaborative reflection among the co-authors, inspired by the reviewer’s comments. These arguments were intended to provide deeper insights into our reasoning and future directions, rather than to disregard the reviewer’s suggestions.
>
> We sincerely hope this clarification underscores our respect for the reviewer’s input and our shared commitment to advancing the quality of this work. Thank you again for your valuable time and efforts.

---

### Official Review · Reviewer_7JuN · 2024-11-02

**Soundness:** 3
**Presentation:** 4
**Contribution:** 3
**Rating:** 8
**Confidence:** 4

**Summary:**

The paper presents a method for obtaining high-level actions from an offline (state only) dataset, by solving the lower level policy inverse problem. The motivation is very clear: High-level, temporal abstractions and hierarchical decomposition is crucial for solving many complex, long horizon problems, therefore, a method for extracting high-level options from an offline dataset that does NOT contain those high-level actions is crucial. In addition, the paper propose to exploit the extracted high-level actions for performing offline reinforcement learning with the resulting high-level dataset, such that the found high-level policy can parameterize the low-level policy at inference time. The proposed method (OHIO) is evaluated on diverse environments (multiple robotics problems as well as network optimization), and performs favorably in comparison with numerous online and offline RL methods.

**Strengths:**

Very well written, clearly motivated, thoroughly validated both empirically and analytically.
To the best of my knowledge this work addresses an important gap in the literature, the possibility to learn high-level actions that parameterize arbitrary low-level policies from state-only trajectories (and assuming approximate knowledge about transition dynamics).

**Weaknesses:**

Not much here to be honest. The only thing that comes to mind is that it was not immediately clear to me how to solve Eq (6) in practice and in the general case, but this becomes very clear with Appendix A.4 and algorithm 2 and 3. Perhaps the content of A.4 could be more integrated into the main method section.

It seems to me like method mainly deals with the implicit policy case, therefore, I am not sure how much value is being added by discussing the explicit policy case.

**Questions:**

How much is this method affected by the quality of the approximate transition model? It would be extremely interesting to see how well the method performs when the ground truth transition model is used, compared with a model learned from the offline data (I guess this would only be possible when low-level actions are part of the trajectories). Did one of the experiments somehow access this and I missed it?

---

> ### Author Response · Authors · 2024-11-21
>
> We thank the reviewer for their comments. Below, we respond to each comment/question individually.
>
> ***Perhaps the content of A.4 could be more integrated into the main method section.***
>
> We thank the reviewer for the positive assessment of our work. Upon reflection, we agree that the discussion on numerical low-level policy inversion can be improved and we incorporated elements of Appendix A.4 to the main body of the paper in Section 3.2.
>
> ***It seems to me like method mainly deals with the implicit policy case, therefore, I am not sure how much value is being added by discussing the explicit policy case.***
>
> Thank you for pointing out this lack of clarity in the current version of the manuscript. Specifically, explicit policies are considered in the robotic experiments, either through LQR controllers (Section 5.1.1) or operational space controllers (Section 5.1.2).
> In light of this discussion, we have revised Section 5 to clearly emphasize this distinction within the main body of the manuscript.
>
> ***How much is this method affected by the quality of the approximate transition model? It would be extremely interesting to see how well the method performs when the ground truth transition model is used, compared with a model learned from the offline data (I guess this would only be possible when low-level actions are part of the trajectories). Did one of the experiments somehow access this and I missed it?***
>
> The reviewer raises an interesting point. In the current version of the manuscript, we have not explicitly analyzed how the quality of the approximate transition model influences the inverse method. Using the ground-truth transition model could provide a useful “upper bound” for OHIO’s performance and help quantify errors arising from approximations of the dynamics.
>
> As the reviewer correctly noted, learning a dynamics model requires observations of low-level actions, which is a typical setup in offline RL. In this case, we generally favor policy inversion in the observed-action case, as discussed in Appendix A.1, where OHIO operates without relying on approximate dynamics knowledge. This approach further relaxes the underlying assumptions.
>
> While we believe that an in-depth exploration of the impact of the quality of the approximate transition model—both in the state-only case and when actions are observed—would be a fruitful avenue for future work, we hope the reviewer agrees that this is an orthogonal direction that would require extensive evaluation across different systems and dynamics models.

---

### Official Review · Reviewer_a4kN · 2024-11-04

**Soundness:** 3
**Presentation:** 3
**Contribution:** 2
**Rating:** 6
**Confidence:** 3

**Summary:**

Authors introduce a new offline HRL approach that can recover high level actions given trajectories of state transitions.  To find the most likely high level actions, a new simple maximum likelihood approach is used.  They show their approach is able to solve a series of robotics and network optimization tasks.

**Strengths:**

- The approach to discovering the high level actions given a set of state transitions is simple.  Select the high level action with the highest likelihood.
- Significant empirical outperformance versus baseslines
- Empirical performance was robust to different controllers and modeling errors
- Applied algorithm to domains with very high dimensional action spaces (i.e., the network optimization problems)

**Weaknesses:**

The approach does appear to have significant limitations.
1.  The approach assumes access to an approximate model of the transition dynamics.  The authors argue this can be relatively simple to learn because it only requires a single step, but this can be difficult in high-dimensional and stochastic settings.
2.  The approach often assumes a pre-trained low-level controller.

**Questions:**

1. Can you clarify what the observed state baseline is?  It seems that it is a high-level policy that always selects the next state from a trajectory as the subgoal?  Why was the next state chosen instead of e.g., 10 timesteps later?  Can you provide some intuition on why is performed poorly?
2.  Can you provide some data on the time horizon (i.e., number of primitive actions) to complete each task?

---

> ### Author Response · Authors · 2024-11-21
>
> We thank the reviewer for their comments. Below, we respond to each comment/question individually.
>
> ***The approach assumes access to an approximate model of the transition dynamics. The authors argue this can be relatively simple to learn because it only requires a single step, but this can be difficult in high-dimensional and stochastic settings.***
>
> We thank the reviewer for the opportunity to elaborate on a core aspect of our work.
> Specifically, rather than proposing a framework that always relies on approximate knowledge of the system dynamics and reward function, our work focuses on highlighting the benefits of using this system knowledge whenever it is available. Concretely, our work considers two extremely widespread (and economically critical) scenarios where some elements of domain knowledge are intrinsically available:
> - Manipulation tasks (e.g., RoboSuite): Having access to the arm’s dynamics model is common practice and necessary for running a lower-level controller. We do not assume knowledge about the stochastic parts of the environment, such as the dynamics of objects. Notably, almost all real-world robotic systems rely on some low-level domain knowledge to ensure the effectiveness of the low-level controllers.
> - Real-world (network) decision systems: Operators typically possess a sufficient degree of actionable domain knowledge. For example, operators might aim to maximize known metrics such as profit or welfare, which are usually easily accessible and well-defined. Similarly, real-world decision algorithms within these systems traditionally rely on deterministic approximations of the dynamics, such as simple macroscopic approximations derived by traffic flow theory.
>
> Moreover, in our experiments on the DeepMind Control Suite (Section 5.1), we explicitly address the setting in which we do not assume specific domain knowledge or dynamics approximations and learn the dynamics model.
> Finally, as discussed in Appendix A.1, in scenarios where low-level actions are observed—arguably the most common setup in offline RL—OHIO does not rely on approximate dynamics knowledge, further relaxing these assumptions.
>
> In summary, we emphasize the following points:
>
> - OHIO is not inherently limited by the assumption of having access to specific domain knowledge. Rather, we discuss actionable ways to greatly improve the performance and applicability of standard offline RL methods by leveraging potentially available domain knowledge.
> - The type of domain knowledge we assume in our experiments is extremely common within a broad range of applications. Although we focus on robotic manipulation and network optimization problems in this work, OHIO could easily be applied more broadly.
> - When low-level actions are observed (i.e., the standard setup in offline RL), these assumptions can be further relaxed.
>
> ***The approach often assumes a pre-trained low-level controller.***
>
> The reviewer rightly highlights OHIO’s compatibility with pre-trained (learning-based) controllers. However, a key strength of our framework lies in its ability to leverage non-learning-based controllers, which are by far the most commonly used in the application areas we consider (e.g., operational space controllers for manipulation tasks).
> Notably, OHIO’s ability to learn hierarchical policies within “classical” (non-learning-based) policy structures enables several advantages, for example:
> - Learning high-level policies over convenient state representations (e.g., operational space instead of joint space for robotic manipulation)
> - Combining the strengths of learning- and non-learning-based methods, such as integrating safety guarantees into learning-based policies.
>
> In summary, while we do not directly incorporate learned (or pre-trained) controllers in this work, our framework is amenable to any kind of low-level controller. In the revised manuscript, we explicitly clarify the assumptions introduced by OHIO across the different settings.

---

> ### Author Response · Authors · 2024-11-21
>
> ***Can you clarify what the observed state baseline is? It seems that it is a high-level policy that always selects the next state from a trajectory as the subgoal? Why was the next state chosen instead of e.g., 10 timesteps later?***
>
> Thank you for giving us the opportunity to clarify the Observed State baseline, which serves as a direct ablation of OHIO’s inverse optimization process.
>
> Specifically, given the same dataset of low-level trajectories (i.e., $\tau = \{s_0, a_0, s_1, a_1,...\}$) OHIO and “Observed State” differ solely in how they reconstruct the high-level trajectories (i.e., $\tau = \{s_0, u_0, s_N, u_N, ...\}$), where $N$ is a hyperparameter defining the temporal abstraction introduced by the hierarchical policy.
> While OHIO solves an inverse problem to recover the high-level action that most likely generated the observed trajectory, the Observed State method bypasses this process by directly selecting observed goal states from the low-level dataset (i.e., $u_t = s_{t+N}$). Crucially, in our experiments, “Observed State” and OHIO share the same (i) temporal abstraction, (ii) goal state representation, and (iii) low-level controller, thus clearly isolating the contribution of the inverse optimization process.
>
> ***Can you provide some intuition on why it performed poorly?***
>
> To better discuss the intuition behind the poor performance of the “Observed State baseline”, let us revisit the example in Appendix C.1 (Figure 5) in the revised manuscript. In this experiment, OHIO solves the inverse problem for a lower-level LQR policy by applying Equation 9, as discussed in Example 3.2. Specifically, the high-level action (i.e., $u_t$) leading to the observed state (i.e., $s_{t+N}$) is computed as a function of the low-level controller parameters (i.e., Q and R in the LQR policy).
> Importantly, the results demonstrate that for varying Q and R, the high-level action leading to the observed state is never the observed state itself. Instead, it depends on the low-level controller’s ability to reach the goal state. Consequently, any high-level policy trained to set observed states as goal states will fail to recover those states in practice.
> While this limitation is less problematic for online RL methods—where online interaction allows the high-level policy to “correct” for such discrepancies—it is critical in offline learning, whereby the accurate high-level trajectory reconstruction enabled by OHIO becomes essential for success, resulting in a clear advantage over the Observed State baseline.
>
> ***Can you provide some data on the time horizon (i.e., number of primitive actions) to complete each task?***
>
> In what follows, we provide the time horizon (and corresponding number of primitive actions) needed to complete each task.
>
> *Goal-directed control*: this task involves 1,000 low-level and 200 high-level actions. The objective is to reach the target as quickly as possible. For an expert policy, this is typically achieved within approximately 50 low-level actions.
>
> *Manipulation task - Lift task*: this task involves 2,500 low-level and 500 high-level actions. The objective is to lift an object and maintain it above a certain height for as long as possible within the given timeframe.
>
> *Manipulation task - Door opening task*: this task involves 2,500 low-level and 500 high-level actions. The objective is to open the door as wide and as quickly as possible. An expert policy can complete the task in approximately 80 high-level actions (equivalent to 400 low-level actions).
>
> *Network optimization:* In the vehicle routing scenario, the high-level policy operates at 3-minute intervals, with each episode consisting of 20 time steps, corresponding to a total duration of 60 minutes. The low-level policy is a network flow problem, where low-level actions do not have a pre-defined temporal dimension rather are assumed uniform within the 3-min discretization. In the supply chain scenario, actions are taken at a daily frequency, with one episode spanning 30 days.
>
> Generally, both network optimization scenarios are inherently infinite-horizon tasks and are not designed to have a defined goal, which marks the task as completed.

---

### Author Response · Authors · 2024-11-21
**General comment - rebuttal summary**

We sincerely thank the reviewers for the many constructive suggestions and comments. We hope that our responses can be a starting point for further discussion within the time limits of this rebuttal phase. We are currently working on a revised paper draft based on comments both on the writing and additional experiments.
While the reviews touched on many important aspects of this work, in our view, the main discussion points of this rebuttal can be summarized as follows:

***1. Clarifying misunderstanding about the need for domain knowledge***:

This rebuttal addresses a misunderstanding regarding OHIO’s reliance on specific domain knowledge, which was highlighted as a potential weakness of the framework. Crucially, rather than proposing a framework that *always* relies on approximate knowledge of the system dynamics and reward function, our work focuses on highlighting the benefits of using this system knowledge *whenever it is available*.
In particular, we validate OHIO across several scenarios where domain knowledge is naturally available. For example, having access to a robotic arm’s dynamics model is a common and necessary practice for performing manipulation tasks. These types of scenarios are highly prevalent in real-world applications and far outnumber cases where system knowledge is unavailable.
Finally, as discussed in Appendix A.1, in scenarios where low-level actions are observed—arguably the most common setup in offline RL—OHIO does not rely on approximate dynamics knowledge, further relaxing these assumptions.

***2. Additional Experiments: Imitation learning from expert demonstrations***:

To better address comments from reviewers, we conducted additional experiments comparing OHIO to popular hierarchical imitation learning methods, such as [2], across multiple datasets. While learning from a single expert dataset proved challenging due to limited data coverage, we addressed this challenge by merging demonstrations from multiple expert policies. Our results demonstrate that OHIO effectively integrates diverse data sources, outperforming alternative approaches.

***3. Writing improvements to the revised manuscript***:

Based on reviewer feedback, we have incorporated the following changes:

- Additional baselines: Appendix C3 now includes additional experiments that compare OHIO to a broader set of baseline methods.
- Integrated elements from Appendix A.4: Key insights on numerical policy inversion have been moved to the main body to enhance clarity and accessibility.
- Explicitly specified policy usage in Section 5: We have clarified the distinction between explicit and implicit policies to ensure better understanding of their respective roles.
- Enhanced Example Appendix C.1 (Figure 5): Updates provide improved intuition regarding the limitations of the “observed state” baseline, emphasizing its impact on performance.
- Refined Algorithm 1: Additional details have been included to better illustrate the steps in OHIO’s framework and address ambiguity concerns raised by reviewers.

[2]. Relay Policy Learning: Solving Long-Horizon Tasks via Imitation and Reinforcement Learning (Abhishek Gupta, Vikash Kumar, Corey Lynch, Sergey Levine, Karol Hausman)

---

### Meta-Review · Area_Chair_hvcT · 2024-12-19

**Metareview:**

The paper proposes an inverse method for offline RL using hierarchical policies. The paper's rating from the reviewers is borderline. The main concerns are the need of approximate dynamics model and the lack of baselines. After examining the discussion, I think the contributions of this paper outweigh the concerns. Nonetheless, I would suggest the authors to highlight the limitation on dynamics model in the revision, as it does exclude certain use cases, and incorporate additional open-source baselines. Also, while not raised by reviewers, I suggest authors add more discussion on the assumption on data quality and property (e.g., for the case of known and unknown approximate dynamics model), as the data assumption is central to offline RL.

**Additional Comments On Reviewer Discussion:**

Reviewer a4kN raises concerns about access to approximate dynamics model, and pretrained low level controller. The authors give arguments that dynamics model info is often available and point out experiments where dynamics models are learned. They also clarify this need is due to the lack of low-level action. I agree with the reviewer that the need of approximate dynamics model as a requirement can weaken the applicability of this method (e.g., while robot dynamics are known for free space reaching, but the contact dynamics like for grasping is non-trivial to model). Nonetheless, I can understand this requirement is necessary duo the nature of the proposed method. I think this limitation needs to be highlighted and acknowledged so that readers can understand when this method can be applied. However I don't think this limitation would constitute to the rejection, as there're also many model-based algorithms which make similar assumptions.

Reviewer 7JuN raises minor questions on clarity. They're addressed during rebuttal.

Reviewer dYhu raises concerns on weakness in experiments, missing comparison to SoTA methods, and clarity on writing. Most of the concerns are addressed during discussion. However, Reviewer dYhu still suggests the need of comparison with more SoTA methods whose code is open-sourced.

---

### Decision · Program_Chairs · 2025-01-22

Accept (Poster)